# Immersive Extended Reality (I-XR) in Medical and Nursing for Skill Competency and Knowledge Acquisition: A Systematic Review and Implications for Pedagogical Practices

**DOI:** 10.3390/bs15040468

**Published:** 2025-04-04

**Authors:** Jennifer M. B. Fugate, Michaela J. Tonsager, Sheila L. Macrine

**Affiliations:** 1Department of Health Service Psychology, Kansas City University, Kansas City, MO 64106, USA; 2College of Osteopathic Medicine, Kansas City University, Kansas City, MO 64106, USA; michaela.tonsager@kansascity.edu; 3Department of STEM Education, University of Massachusetts-Dartmouth, Dartmouth, MA 02747, USA; smacrine@umassd.edu

**Keywords:** medical/healthcare education, conventional simulation-based learning, Immersive Extended Reality (I-XR), learning theory

## Abstract

Simulation has evolved from basic practice to Immersive Extended Reality (I-XR). This systematic review examined 56 published studies on the impact of I-XR, including virtual reality (VR), augmented reality (AR), and mixed reality (MR), on the education of medical and nursing students, specifically their skill competency, and knowledge acquisition. The results demonstrate the significant potential of I-XR in healthcare education, with 42.5% of VR studies, 42.9% of AR studies, and the single MR study also demonstrating greater improvements in clinical skills and knowledge acquisition compared to non-immersive (non-I-XR) training conditions. In contrast, only 2.5% of VR studies and 7.14% of AR studies favored non-I-XR methods. It is important, however, to acknowledge the 26.8% of studies that showed mixed results (some evidence for the I-XR methods on some outcomes, but also some evidence for the non-I-XR methods, on other outcomes). Notably, the review also identified a critical gap in the theoretical foundations of I-XR learning, highlighting the urgent need for research to inform the effective pedagogical implementation of these powerful tools. We offer a preliminary framework to address the lack of learning theory in healthcare I-XR training, with implications for pedagogical practices.

## 1. Introduction

Simulation-based training is a pedagogical approach that employs simulated scenarios or environments to improve clinical skills, mainly among health field workers. The objective is to replicate real-world situations to provide a realistic and immersive experience without exposing patients to real risks ([16]). Simulation-based training complements conventional teaching methods by focusing on improving skill acquisition, decision-making, and teamwork ([4]). Such clinical experiences can help create opportunities for students to apply didactic content to clinical practice, which allows them to bridge knowledge, skills, and competency ([46]).

Halsted’s “see one, do one, teach one” model, while groundbreaking in 1890, is now criticized for prioritizing rapid experience over deliberate skill development and patient safety ([9]; [14]). This approach raises concerns about inadequate preparation, potentially leading to medical errors and a lack of emphasis on critical thinking and self-reflection ([99]). Modern medical education is shifting toward comprehensive training models that prioritize patient safety and incorporate simulation, deliberate practice, and feedback ([9]; [14]).

Simulation-based medical education (SBME) ([102]) offers a viable solution for bridging this gap. SBME enables deliberate practice and standardized training in a safe learning environment. It incorporates a wide array of techniques, ranging from basic mannequin-based simulations and standardized actor-patients to complex multimedia scenarios ([105]; [132]). Through SBME, students can build confidence and competence and ultimately enhance patient safety and the overall quality of care.

According to [14] ([14]), SBME has three primary objectives: (1) the execution of clinical skills, (2) the supervised practice of that skill, and ultimately, (3) the independent and confident performance of clinical skills. This educational approach extends beyond technical proficiency to encompass the development of exteroceptive awareness among aspiring professionals. Exteroceptive awareness, or the ability to perceive and interpret external stimuli, is crucial for stress management, confidence building, and overall well-being ([14]). Specifically, cultivating exteroceptive awareness helps regulate the sympathetic nervous system and manage stress responses in high-pressure situations, such as surgical procedures, thereby enabling healthcare professionals to maintain their composure and perform optimally.

Specifically, significant positive effects of SBME (vs. non-simulated methods) have been shown for theory knowledge, analytic skills, learning interest and understanding, satisfaction, cooperative ability, problem-solving ability, teaching success, and situation awareness ([111]). Additionally, SBME is effective in assessing teamwork and communication among healthcare providers ([33]; [104]). These benefits translate into improved patient outcomes and reduced healthcare costs ([72]; [49]), establishing it as a fundamental pillar of healthcare clinical training ([102]) that provides a safe and effective environment for learners to develop crucial skills and decision-making abilities ([95]).

### 1.1. Integration of I-XR in Healthcare Education

Recent technological advancements have further propelled SBME to include methods like virtual reality (VR), augmented reality (AR), and mixed reality (MR) ([113]). These methods are thought to better equip healthcare students with practical experience and readiness for actual scenarios and procedures ([43]; [125]). Such technologies are increasingly seen as having promise for teaching healthcare professions, even though the largest market is still in the entertainment and gaming industry ([12]; [123]).

Extended reality (XR) is an umbrella term used to include VR, AR, and MR ([2]), but typically refers to computer-generated images in the wearer’s field of vision and includes the range of the user’s view of the world that can exist from fully visible to fully occluded ([130]), thus blurring the line between the digital world and the physical world ([30]). Finally, Immersive Extended Reality (I-XR) is characterized as a screen-based simulation that highlights the 3D nature of patients, graphics, sound, and navigation through the environment ([83]). The use of I-XR for healthcare education is becoming increasingly recognized as it allows an almost unlimited number of clinical scenarios to be simulated, with the ability to allow real-time feedback on student progress and patient status ([77]), and can improve users’ motivation, engagement, and enjoyment in educational learning across different domains by providing powerful experiential learning ([35]; [114]). As more healthcare educators pioneer innovative methods, I-XR is emerging at the forefront of teaching and learning and is preparing healthcare students with practical skills, hands-on experience, and preparedness for real-world scenarios and procedures ([43]). Embracing these changes represents an innovative step toward cultivating a more knowledgeable, proficient, and self-assured healthcare workforce, benefiting patients and enhancing healthcare education.

### 1.2. Review of I-XR Studies for Training Effectiveness in Healthcare Education

Whether I-XR methodologies specifically improve student learning outcomes compared to non-immersive (non-I-XR) methods remains a subject of debate. Historically, the majority of meta-analytic studies have focused on medical students’ surgical skills ([130]). Within the last five years, there has been a surge of systematic reviews and statistical meta-analyses seeking to explore whether I-XR (specifically VR, AR, and MR) methodologies improve students’ learning outcomes across domain-specific skills, procedural outcomes, and even non-technical skills (e.g., empathy, self-efficacy, and teamwork). Despite the number of reviews, the results are still inconclusive and often depend on how “knowledge acquisition” and “skill competency” are operationalized. For instance, a meta-analysis of 11 studies on VR endoscopy training for medical students found that while VR improved procedure completion and overall performance ratings compared to traditional methods, it resulted in fewer independent completions and showed no difference in other behavioral outcomes ([62]). Another VR systematic review of nine studies revealed that only two showed improved knowledge acquisition in healthcare professionals compared to the control groups ([1]), suggesting that while VR can enhance knowledge, it may not be consistently superior to other methods. However, the same review found significant increases in skill competency in 19 out of 21 studies comparing VR to control methodologies ([1]), indicating a stronger impact on practical skills. Other recent reviews of VR technology ([45]; [51]; [115]) have shown the positive effects of I-XR on nursing students’ knowledge, skill performance, skill acquisition, *and* clinical reasoning. These varied results suggest that the effectiveness of I-XR may depend on factors such as the specific technology used, the type of training, and the outcome being measured. Further research is needed to fully understand the conditions under which I-XR can optimize learning outcomes in healthcare education.

The results are also mixed as to whether one type of immersive technology (AR, VR, or MR) is more effective than another. For example, a systematic review of VR head-mounted displays in medical education ([67]) showed significant increases in learning surgical *procedures* in seven of the 11 studies reviewed compared to other immersive methodologies, but no differences were observed when directly compared with AR. However, when assessing anatomy *knowledge*, only one of the six studies reviewed showed increased anatomy knowledge using VR. The authors concluded that VR technologies outperformed conventional methods for learning surgical skills, but they were not superior for learning anatomy knowledge. However, a recent meta-analysis ([28]) demonstrated that VR-based methodologies were significantly more effective than traditional methods, such as non-VR simulation or didactic instruction, in enhancing knowledge acquisition among nursing students. Notably, this effect was specific to knowledge acquisition and did not extend to performance time or skills. Despite mixed evidence for I-XR technologies compared to non-immersive technologies in learning procedural skills and knowledge acquisition, specific teaching and learning processes remain underexplored. Therefore, how these methods use or adhere to learning theories to understand the delivery of skills and knowledge could be another overlooked possibility for mixed evidence.

For example, educational learning theories offer valuable frameworks for understanding how students acquire knowledge and develop skills. Researchers have identified 13 key theoretical perspectives relevant to medical education, including cognitivism, constructivism, experiential learning, and reflective learning ([61]; [81]). These theories guide a range of pedagogical approaches, from optimizing information presentation ([48]) to fostering reflective practice ([116]). However, [97] ([97]) found that most studies in the healthcare field lack a foundation in learning theory, with many failing to support specific learning processes or explicitly mention any theoretical framework.

However, despite the potential benefits of integrating learning theories into I-XR research, several studies have identified concerning trends. Many studies on I-XR in healthcare education either poorly described or completely omitted their pedagogical approaches ([63]). In one study designed to specifically assess whether VR studies incorporated a conceptual framework or theory in their design, less than 3% of papers on VR simulations did so ([54]). This lack of theoretical grounding raises concerns regarding the rigor and generalizability of I-XR research in healthcare education. Without a clear theoretical foundation, it is difficult to understand why certain interventions are effective and how they can be optimized for different learning contexts. This highlights the critical need for researchers to explicitly integrate learning theories into the design, implementation, and evaluation of I-XR training programs.

### 1.3. Summary

In summary, I-XR is transforming healthcare education by providing immersive and engaging learning experiences that enhance skill acquisition, knowledge retention, and preparedness for clinical practice. This technology aligns with the evolving needs of healthcare training, offering adaptable and flexible solutions for integrating cutting-edge techniques. As technology continues to advance, I-XR is poised to play an increasingly vital role in shaping the future of healthcare education.

While learning theories offer valuable insights into teaching and learning processes, their application in healthcare education research remains surprisingly limited. This gap is concerning because a strong theoretical foundation is essential for maximizing the pedagogical impact of any educational intervention. The omission of learning theory is particularly critical in the context of I-XR, where the novelty and complexity of the technology necessitate a strong pedagogical underpinning to guide its effective implementation. To realize the full potential of I-XR in healthcare education, it is crucial to equip healthcare educators with the necessary skills and resources to effectively integrate these technologies into high-quality patient care training. This includes providing professional development opportunities that focus on applying learning theories to the design and delivery of I-XR-based learning experiences.

### 1.4. Purpose

This systematic review aimed to examine the effectiveness of I-XR training for healthcare students (e.g., medical and nursing students). We compared I-XR training methods to traditional, non-immersive approaches to determine whether I-XR leads to greater improvements in skill competency and knowledge acquisition. These competencies were assessed across multiple domains, including performance skills, objective performance measures, clinical reasoning, internship grades, problem-solving skills, and skills knowledge This review has significant implications for the healthcare field. First, it examines six distinct outcomes related to skill competency and knowledge acquisition. Second, it focuses on both graduate medical and nursing education across a wide range of techniques, offering a holistic view of I-XR’s potential in healthcare training. Third, recognizing that the effectiveness of I-XR may vary depending on the pedagogical approaches to which they are compared, the review analyzes the results according to the type of non-immersive (comparison) group used (e.g., didactic instruction, manikin-based training, simulated patients). This nuanced approach provides valuable insights into the specific contexts in which I-XR excels. Fourth, unlike many reviews that focus on a single technology, this review encompasses the full breadth of I-XR techniques, including AR, VR, and MR allowing for a comprehensive understanding of the diverse applications and benefits of I-XR in healthcare education. Finally, and perhaps most importantly, it assesses each study for the incorporation of learning theory, enabling an evaluation of whether skill competency and knowledge acquisition are grounded in established pedagogical techniques. This critical analysis highlights a significant gap in the current research and underscores the need for a stronger theoretical foundation for I-XR implementation. To address this gap, in the discussion section, we offer a preliminary framework for integrating embodied learning theory into healthcare I-XR training, with implications for pedagogical practices. This framework provides a valuable tool for educators and researchers seeking to maximize the pedagogical effectiveness of I-XR in healthcare education.

## 2. Methods

### 2.1. Search Strategy

Consistent with the Best Evidence in Medical Education (BEME) recommendations, the search was performed on the inclusive databases of PubMed and Google Scholar in January 2023 and again on 1 April 2024, following the Joanna Briggs Institute for scoping reviews. The search terms included “simulation in medical (or nursing) education” and “augmented reality (AR)” [or virtual reality (VR)” or “mixed reality (MR)”] and “skill outcome” (or “skill performance”). We selected articles based on the PRISMA method (see Figure 1). We have adhered to the PRISMA guidelines. This review was not pre-registered.

### 2.2. Inclusion and Exclusion Criteria

Only peer-reviewed empirical articles written in English between the dates of 1 January 2016 and 1 April 2024 were included. XR papers surged in 2017; thus, this window captures the increase from 2016 to the present ([117]). Dissertations, letters to editors, commentaries, and opinion pieces were excluded. Only articles with a control group were included (i.e., a within-subject pre-post-test design or an external comparison group). We also included only articles that focused on objective performance measures of skill competency and knowledge acquisition (i.e., performance skills, internship grades, performance measures, clinical reasoning or judgment, and problem-solving skills). Lastly, we included only articles that included graduate medical or nursing students. Studies were excluded if they were outside the indicated dates, did not include a control group or pre-post design, did not include medical or nursing graduate students (e.g., undergraduates), were not available in English, or were not published in a peer-reviewed journal. Studies that solely included measures of confidence, self-esteem, or other non-technical skills were excluded from the process. For cases in which the studies included these measures but also included skill competency and/or knowledge acquisition measures, we reported only the latter.

The initial search with the parameters yielded 11,434 articles. Duplicates were removed, and the titles and abstracts were evaluated for topic appropriateness. We also reviewed additional papers using the ancestral method (i.e., reviewing the references of the included papers) that did not appear in our original search. After screening, 156 papers remained for review (Figure 1). After reading these papers in full, 56 were included in the final data set (*n* = 100 removed for reasons indicated above) (Appendix A).

### 2.3. Data Extraction

We extracted the year, student sample size, field (e.g., medical and nursing), type of I-XR (AR/VR/MR), comparison group type, outcome measures, and any mention of learning theory. We also included a description of each study’s methods, results, and descriptive statistics for comparisons of interest (Table 1). At least three researchers reviewed each paper for the accuracy of the variables. Two coders coded the year, student sample, field, learning theory, and type of I-XR method used. The agreement was >95% on the first pass. The first and second authors coded the types of comparison groups and outcome measures and made the final assessment for each. The initial agreement exceeded 85%. If the codes were discrepant, the first author made the final decision.

### 2.4. Data Coding

Outcome measures (i.e., skill competency and knowledge acquisition) were initially operationalized into six categories based on past literature: “performance skills, including “direct observation of procedure” (DOPS); “performance measures” (Objective Structured Clinical Exams (OSCE), Global Operative Laparoscopic Assessment (GOALS), Global Rating Scale (GRS), Academic Achievement Test (AAT); “clinical judgment or clinical reasoning” (Laster Clinical Judgment Rubric (LCJR); “skills knowledge” (e.g., Mini-Clinical Evaluation Exercise (CEX), Mini-CEX, Neurological Physical Exam (NPE), multi-choice questions of skills (MCQ); “grade in internship”; and “problem-solving skills”. We then felt it necessary to subdivide “performance skills” into specific modifiers, regardless of the actual behavior (e.g., “performance skills errors”, “performance skills time”, “performance skills injury”, “performance skills dexterity”). Finally, we added an additional category: “performance skills other”, for cases in which the performance skill was not one of those noted above. If a study had more than one outcome measure, we evaluated each outcome separately. Therefore, the total number of outcomes exceeded that of the studies.

The comparison groups were divided into seven categories based on previous studies. These groups included learning with: “print materials” (e.g., study guides, books, technical manuals), “teacher-led” (e.g., didactic/instructor-led), “electronic materials” (e.g., video tutorial, e-learning/computer materials), “practice” (e.g., BOX trainer, dissection, mannikin, simulated patient, case-based learning (CBL), or hands-on with instructor that was not immersive-XR). If a study had multiple comparison methods (e.g., electronic materials and practice), we coded it as “combined”. However, if a study included some students learning through print materials (comparison 1) and some learning via practice, the study was evaluated twice, in this case—once for each comparison group. If *all* students had received additional training of some sort (including the I-XR group), we did not include that training as a comparison group. In these cases, the comparison group was coded as “did nothing additional”. Finally, some studies were pre-post studies with a comparison group. In these cases, the results were assessed concerning differences between the I-XR group and the comparison group(s) *after* training rather than any pre-post change.

We gave each study one of five final assessments (Table 1): (1) “Positive” (total support for I-XR methods on all outcomes for all comparisons), (2) “Negative” (total support for the control methods on all outcomes for all comparisons), (3) “No difference” (I-XR and control methods produced no differences on all variable and comparisons), (4) “Mixed Evidence Positive” or (5) “Mixed Evidence Negative”. “Mixed Evidence Positive” was used for cases in which the I-XR methods produced enhanced effects on more outcomes compared to the non-immersive (comparison) methods. Similarly, “Mixed Evidence Negative” was used for cases in which the comparison methods produced enhanced effects on more outcomes compared to the I-XR methods. Thus, “Mixed Evidence Positive” was given to any study that favored the I-XR methods, even if on some of the variables, there were no differences with the comparison, or there were some differences that favored the comparison methods (as long as that number was fewer than those favoring the I-XR methods). A similar logic was used for “Mixed Evidence Negative”, except that the evidence favored the comparison methods. There was only one instance in which “Mixed” without a qualifier was used. This is because there were two outcomes, and for one, the I-XR methods were superior, whereas for the other, the comparison method was superior.

### 2.5. Assessing Article Quality and Bias

There are several ways in which article quality and bias can be addressed, including the [31] ([31]) instrument ([27]), the Mixed Methods Appraisal Tool (MMAT) ([42]), the [55] ([55]), the Medical Education Research Study Quality Instrument (MERSQI) ([52]), the Newcastle-Ottawa Scale for Education (NOS-E) ([119]), and the Quality Assessment of Diagnostic Accuracy Studies revised (QUADAS-2) ([120]). In our exploration of the literature, we found that the most common assessments were MERSQI, NOS-E, and QUADAS-2. Because both the MERSQI and the NOS-E evaluate different aspects of study design and quality, we chose to evaluate each study on both.

The MERSQI is used as a measure to assess the quality of studies across eight domains: (1) study design, (2) sampling, (3) response rate, (4) type of data, (5) validity of evidence for evaluation measures, (6) data analysis sophistication, (7) data analytic appropriateness, and (8) outcome. The maximum score for study design, validity of evidence, outcome, and type of data is “3”. For sampling design and response rate, the maximum score is “1.5”. For data analysis sophistication, the maximum score is “2”, and for data analysis appropriateness, the maximum score is “1”. The scores are summed, with a maximum score being 16. We also included items from the NOS-E, which evaluates a study on sample representation, comparison groups, retention, and blinding conditions. Sample representation, selection of the comparison group, retention, and blinding conditions were all scored with a maximum of “1”, whereas the comparability of the comparison group was divided into randomized and non-randomized studies, each having a maximum score of “2”. The scores are summed, with a maximum score being eight. The maximum score for the total of both scales was thus “24”. The average MERSQI-2 plus NOS-E score across the 56 studies was 18.5, showing a generally high quality of articles included (range: 15–22) (see Table 2).

To assess bias, we chose QUADAS-2. QUADAS-2 includes 11 items that assess the risk of bias with “yes”/“no”/“unclear” marks. We included a total “yes” score that was summed across the 11 statements, indicating bias. The average QUADAS-2 score across studies was 7.12 (range: 2–11), showing a moderately high bias despite the high study quality (see Table 2).

## 3. Results

### 3.1. Descriptive Findings

Of the 56 articles reviewed, 40 (71.43%) were VR studies, 14 were AR (25.0%), one study was an MR study, and one study was a VR/AR study because it was unclear whether the training was VR or AR based on the description of the apparatus (1.8%) (see Figure 2).

Most studies were published in 2020 (n = 13, 23.2%) (Figure 3).

A total of 80.4% (n = 45 studies) were conducted on medical graduate students (n = 3641 individuals), compared to 19.6% of studies (n = 11) that were conducted on nursing graduate students (n = 1085 individuals) (Figure 4).

The types of comparison techniques used across studies included “print” (n = 13, 21.0%), “teacher” (n = 8, 12.9%), “electronic” (n = 10, 16.1%), “practice” (n = 15, 24.2%), “combined methods” (n = 6, 9.7%), “pre-post” (n = 3, 4.8%), or “did nothing additional” (n = 7, 11.3%) (Figure 5).

“Performance skills” variables across studies included: “time”, n = 13, 11.3%; “errors”, n = 5, 4.4%; “dexterity’, n = 0, 0%; “injury”, n = 1, 0.9%; “other specific/DOPS”, n = 46, 40.0%) “skills knowledge’ (n = 28, 24.4%), “grade in internship” (n = 1, 0.9%), “performance measures” (n = 15, 13.0%), clinical judgment/reasoning skills” (n = 5, 4.4%), “problem-solving skills” (n = 1, 0.9%) (Figure 6). The average number of variables assessed in a study was two (range: 1–12 variables).

Table 2 also shows whether any learning theories were mentioned in the included studies. Approximately seventy-five percent of the studies did not mention any learning theory. Thirteen studies included one or more theories. The most commonly mentioned theory was the cognitive load theory (CLT) (n = 6). While CLT is often used to inform instructional design, it is primarily a framework for understanding how information is processed in working memory rather than a comprehensive learning theory that explains the complexities of knowledge acquisition and skill development. Simulation theory/NLN/Jeffries theory was mentioned three times, and self-regulated theory and Bloom’s theory were mentioned twice. Other learning theories, including the situated learning theory, directed self-regulated theory, simulation-based mastery learning, constructive alignment theory, deliberate practice, Kolb’s theory, and experiential learning theory, were each mentioned once.

### 3.2. Overall Study Assessments of I-XR Skill Competency

There was only one MR study, and one study was coded as AR/VR because the specific technology could not be identified. The one MR (100%) was more effective than the comparison (Positive). One study classified as VR/AR showed no change compared to the control (No Difference).

With these exceptions, however, there was virtually no difference between the proportion of AR and VR studies that showed “Positive” evidence. Overall, we found that 42.5% of studies reported that VR was more effective than comparison (non-immersive) methodologies, and 42.9% of AR studies were more effective than non-immersive methodologies (Figure 7).

There was also virtually no difference between the proportion of AR and VR studies that showed “No Difference”. Overall, we found that 25.0% of VR studies and 21.4% of AR studies showed no difference between the technology and comparison groups (Figure 7).

The proportion of VR studies showing “Mixed Change” was greater than that of AR studies. Overall, we found that 35.0% of VR studies and 14.3% of AR studies showed “Mixed Change” between that technology and the comparison groups (Figure 7).

Finally, the proportion of AR studies showing “Negative Change” (support for control approaches) was greater than the proportion of VR studies (7.14% vs. 2.50%). This may suggest that AR studies are less effective on the outcomes we measured than AR studies, although a direct statistical comparison was not made.

## 4. Discussion

This review provides compelling evidence for the effectiveness of I-XR training in healthcare education. A substantial majority of studies (42.9%) demonstrated that I-XR methodologies led to universally improved outcomes compared to traditional training comparison methods. Figure 7 shows the effectiveness of specific technologies (MR, VR, AR). These percentages reflect improved outcomes on all measures, thus underscoring the robustness of the findings. Only a small minority (3.6%) of studies universally favored comparison (non-immersive) methods. There was only one MR study that showed positive support for mixed reality. In addition, one study was labeled as AR/VR because the methodology used was not clear. This study revealed no difference between the technology and control technology. This limited sample size highlights a critical gap in the literature and underscores the need for caution in generalizing the results.

It is important, however, to acknowledge that 26.8% of I-XR studies showed mixed results (Mixed Positive, n = 10, 17.9%; Mixed Negative, n = 4, 7.1%; Mixed, n = 1, 1.8%). The effectiveness of I-XR is, therefore, not universal and may depend on various factors, such as the specific technology used, the type of training, and the implementation context. Further research is needed to identify the conditions under which I-XR is most effective and to develop evidence-based guidelines for its optimal use.

A significant gap in the theoretical foundation of I-XR teaching and learning approaches was also identified. Only 14 of the 56 papers (25%) reviewed mentioned any learning theory. When “cognitive load theory” was removed, this shrunk to 16%. Of the 14 papers that mentioned any learning theory, five favored I-XR (“Positive”) (35.7%), while only one favored non-immersive approaches (“Negative”) (7.1%). Two papers produced ‘Mixed Positive” evidence (14.3%), and none produced “Mixed Negative” or “Mixed” evidence. These percentages did not significantly deviate from the overall distribution of outcomes across all studies.

The absence of learning theory in 75% of the reviewed papers is a significant concern. This omission risks impeding the development of effective, evidence-based pedagogical practices for I-XR in healthcare education. This suggests a fundamental gap in understanding how to optimize teaching and learning in these novel immersive environments. A stronger theoretical foundation could help optimize the design, implementation, and evaluation of I-XR-based learning experiences, ultimately leading to improved learning outcomes and better patient care.

Without established theoretical frameworks, I-XR implementations are susceptible to ad hoc designs, which hinder systematic development and rigorous evaluation. Simply transferring traditional pedagogical models from physical classrooms to virtual environments proves inadequate, given the unique affordances and challenges inherent in the I-XR. Addressing issues such as maintaining student engagement, fostering collaboration, and ensuring equitable access—challenges documented in recent research ([134])—necessitates a comprehensive reconsideration of pedagogical approaches. To fully realize the transformative potential of I-XR for simulation, interaction, and experiential learning, a deliberate and sustained effort to integrate existing learning theories and develop novel frameworks tailored to immersive digital environments is indispensable.

While teachers and instructors recognize the value of action-based classroom research, a disconnect often exists between theory and practice. This disconnect manifests as a lack of robust theoretical grounding and a tendency to neglect rigorous empirical methodologies ([40]; [66]). Consequently, a gap emerges between classroom instruction and the actual learning experiences of students. This deficiency in informed practice stems from the absence of a clear framework for translating theoretical concepts into practical teaching strategies, in this instance, linking embodied cognition and embodied learning to effective pedagogical approaches.

Experiential learning, which emphasize learning through direct experience and reflection, has proven to be a powerful pedagogical approach ([64]). Kolb’s theory (1984) outlines a four-stage cycle of learning, starting with concrete experiences that are transformed into abstract concepts and testable hypotheses through reflective observation and active experimentation. This cyclical process is particularly relevant to immersive learning environments that encourage both concrete experiential learning and reflective observation ([32]). I-XR with its ability to create realistic simulations and interactive scenarios, aligns with the principles of experiential learning by providing learners with concrete experiences.

Building on this foundation, the embodied learning theory, grounded in embodied cognition, underscores the crucial role of physical experiences and interactions in shaping cognitive understanding ([121]; [107]). This perspective recognizes that the body and mind are inextricably linked, with learning and comprehension occurring through interactions with the physical world, leveraging sensory input and motor actions to acquire information and construct knowledge. Sensorimotor experiences are fundamental to cognition; seeing, hearing, touching, smelling, and even tasting can deepen our understanding and create more memorable learning experiences ([13]; [37]; [133]; [107]). Embodied learning suggests that learning is optimized through active engagement, sensory immersion, and interaction with the surrounding environment ([78]). This approach is supported by neuroscience research, which elucidates the connection between experiential learning and brain activity ([47]) and demonstrates the effectiveness of body-based learning strategies ([17]).

These ideas underscore the interconnectedness of active engagement, sensory immersion, and interaction with the environment to enhance the learning experience. Furthermore, understanding cognition in this context goes beyond being merely “embodied”; it is also “embedded” or situated within a specific context, “extended” beyond individual boundaries through virtual and simulated practices, and “enacted” as part of a dynamic system ([93]; see [79]). As a result, there is a strong connection between experiential learning and embodied cognition’s 4Es: embodiment, embeddedness, extendedness, and enaction ([89]).

Extended cognition, which posits that cognitive processes extend into the physical world and an individual’s interactions with it ([71]), emphasizes cognition as an active and situated process rather than a purely internal representation of knowledge. Thus, cognitive abilities emerge through real-time interactions between the individual and their environment, in which learning is deeply tied to the physical movements, sensory feedback, and spatial awareness of the learner. By creating interactive virtual environments that allow for physical engagement and manipulation, I-XR provides a powerful platform for extending cognition beyond the confines of the individual mind. This aligns with the experiential learning theory’s emphasis on hands-on learning and the reciprocal relationship between the individual and the learning environment ([65]).

Embedded cognition further suggests that knowledge is situated within specific cultural, social, and professional contexts and that learners actively construct this knowledge through environmental interaction ([122]). Users are pedagogically guided to navigate virtual spaces, manipulate medical tools, and respond to simulated patient conditions in real-time. Research supports this, indicating that emphasizing immersive movement congruent with real-life actions in I-XR environments deepens the understanding of clinical scenarios ([74]).

The convergence of experiential learning and 4E cognition provides a robust theoretical foundation for I-XR’s transformative potential in healthcare education. Moving beyond decontextualized classroom learning, a 4E embodied approach to I-XR immerses healthcare trainees in realistic clinical environments, fostering situated learning that is directly applicable to real-world practice. This embodied approach to teaching and learning in I-XR, which acknowledges the integral role of the body in cognition, enhances learning efficacy. Offering concrete experiences, multisensory engagement, practice opportunities, feedback, and collaborative environments, I-XR facilitates deep conceptual understanding, enhances learner engagement, and builds essential skills. The sense of presence and 3D content manipulation inherent in I-XR are crucial for fostering learner agency and embodied interactions ([56]). This is further substantiated by ([56]; [58]) taxonomy for embodiment, which emphasizes sensorimotor engagement, gesture-content congruency, and immersion (see also [57]).

### 4.1. A New Embodied Learning Model

To enhance the pedagogical effectiveness of I-XR in healthcare education, here we briefly introduce a new embodied learning model ([80]) that aligns Bloom’s Taxonomy ([7]) of hierarchical cognitive levels with Shulman’s pedagogical framework ([108]) (see also [98]). This model helps educators guide students through progressively complex cognitive levels, from foundational knowledge to higher-order thinking skills, such as application, analysis, evaluation, and creation, ensuring a comprehensive learning experience. It comprises three interconnected levels of embodied pedagogical knowledge:Embodied Content Knowledge (ECK): A deep understanding of the subject matter gained through embodied experiences and interactions with the world.Embodied Technological Knowledge (ETK): Knowledge and skills related to using tools and technologies in an embodied manner to explore, analyze, and solve problems.Embodied Pedagogical Knowledge (EPK): Ability to design and facilitate embodied learning experiences that bridge learners’ tacit understanding with formal knowledge and practices.

To further elaborate on embodied pedagogical knowledge (EPK), our framework introduces three interconnected phases specifically tailored to the unique demands of embodied learning in healthcare fields. These phases refine and extend the concepts of Shulman’s pedagogical content knowledge (PCK) and technological pedagogical content knowledge (TPACK) ([84]) to encompass the crucial roles of the body, sensory experience, and interaction with the environment in the learning process. They align with the cognitive progression outlined in Bloom’s Taxonomy.

Remembering and Understanding: I-XR can be used to practice interactive simulations and role-playing scenarios that engage learners’ senses and bodily movements, helping them remember and understand key concepts and procedures. For example, learners can practice basic life support skills on virtual patients in an I-XR environment.Applying and Analyzing: I-XR can be used to have learners apply and analyze their knowledge and skills in context. For instance, learners can participate in a virtual surgery simulation in an I-XR environment, where they need to make decisions, solve problems, and perform procedures based on their understanding of anatomy and surgical techniques.Evaluating and Creating: I-XR can be used to encourage learners to evaluate and create new knowledge and solutions in I-XR. For example, learners can design and conduct virtual experiments in an I-XR to test hypotheses, build virtual models of anatomical structures, and develop new surgical tools and techniques in a virtual environment.

By utilizing Bloom’s Taxonomy in conjunction with Shulman’s framework of pedagogical knowledge, this model provides a structured approach to implementing embodied teaching and learning within I-XR environments. This ensures that learning experiences are intentionally designed to engage learners’ bodies and senses, while simultaneously fostering higher-order thinking skills.

Furthermore, a comprehensive assessment system is crucial for capturing the true impact of I-XR on learning and skill acquisition. This system should consider various aspects of embodied learning, such as agency, proprioception, interoception, engagement, affordance, intercorporeality, embeddedness, and extendedness. This embodied approach to assessment not only provides a more accurate picture of students’ understanding but also fosters deeper learning and greater engagement with healthcare content.

In conclusion, by integrating this model, I-XR can be leveraged to create more effective and engaging learning experiences in healthcare based on a theory of embodied learning. This approach not only enhances knowledge and skills but also fosters a deeper understanding of healthcare concepts and practices through active engagement, embodiment, and interaction with realistic, virtual environments.

### 4.2. Limitations

While I-XR technology holds immense promise for revolutionizing healthcare education, several limitations need to be addressed to fully realize its potential. A critical appraisal of the current landscape reveals methodological weaknesses in the existing body of research, notably the prevalence of bias in some published studies. This bias threatens to inflate the perceived effectiveness of I-XR, hindering our ability to accurately assess its impact on students’ learning and clinical skills.

To address these limitations, future research must prioritize rigorous methodologies, including meticulously controlled randomized trials and objective outcome measurements. This will help eliminate bias and generate conclusive evidence of I-XR’s efficacy. Additionally, technological advancements are urgently needed to reduce costs, enhance user experience, and mitigate any adverse physiological effects, making I-XR more accessible.

Furthermore, fostering collaborative partnerships between educators, technology developers, and institutional stakeholders is essential. These partnerships can address logistical challenges, develop best practices for implementation, and facilitate seamless integration of I-XR into healthcare education. By acknowledging and proactively addressing these limitations, we can pave the way for the successful and impactful integration of I-XR, ultimately transforming healthcare education and improving patient care.

### 4.3. Future Directions

The future of healthcare education is poised for a transformative era driven by the remarkable potential of I-XR technologies. However, this transformation hinges on a fundamental shift in pedagogical thinking. Simply deploying I-XR without a robust theoretical foundation risks creating engaging yet ultimately ineffective learning experiences. These technologies, encompassing virtual reality (VR), augmented reality (AR), and mixed reality (MR), offer unprecedented opportunities to create engaging and effective learning experiences that bridge the gap between theoretical knowledge and practical application. To truly leverage I-XR’s potential, we must move beyond technological adoption, prioritize the application of established learning theories, and develop novel pedagogical frameworks specifically tailored for immersive I-XR environments.

The current lack of theoretical grounding in many I-XR implementations in healthcare and general education underscores the urgent need for a more deliberate and research-driven approach. As highlighted by [75] ([75]), evidence for the effective utilization of I-XR in pedagogical frameworks remains limited. Therefore, a key future direction is the rigorous development and validation of I-XR-specific pedagogical models ([75]).

Adopting the embodied learning framework introduced here represents a promising pedagogical approach and a key component of a robust pedagogical strategy for I-XR healthcare education. This framework emphasizes active engagement, sensory immersion, and environmental interaction, which aligns with the inherent capabilities of I-XR. However, merely selecting a pedagogical framework is insufficient; instructors must be thoroughly trained in its application within I-XR environments. Effective integration also necessitates a collaborative ecosystem involving medical education leaders, faculty, and technology developers to foster a shared vision and provide comprehensive faculty training. Standardized methods for educating, assessing, and certifying I-XR instructors are essential to ensure consistent quality in the pedagogical application of I-XR.

While virtual simulation centers are valuable, the future lies in increased accessibility and versatility, with I-XR seamlessly integrating with mobile devices and wearable technology. To achieve this, research must prioritize the development of pedagogical approaches that adapt to diverse platforms and modalities. Future studies should explore robust pedagogical frameworks grounded in learning theories to determine the conditions under which I-XR training is most effective, considering technology type, learning objectives, and learner characteristics.

Problem-based learning, enhanced by virtual patients and dynamic case scenarios, offers a powerful avenue for I-XR applications. However, these scenarios must be carefully designed to align with sound pedagogical principles to ensure that learning translates into improved clinical reasoning and decision-making. As I-XR becomes more accessible, collaborations will drive personalized learning experiences tailored to diverse needs.

The future promises richer and more immersive experiences with haptic feedback and realistic visuals. However, the true power of these technologies lies in their integration with sound pedagogical strategies. By grounding I-XR experiences in learning theory, we can create immersive and impactful environments that optimize teaching, learning, and evaluation. Crucially, we must avoid the ad hoc design that results from a lack of theoretical guidance, ensuring that I-XR enhances rather than replaces sound pedagogical practice. This commitment to pedagogical excellence, coupled with ongoing innovation and collaboration, will ensure that healthcare education remains at the forefront of technological and educational advancement.

## Figures and Tables

**Figure 1 behavsci-15-00468-f001:**
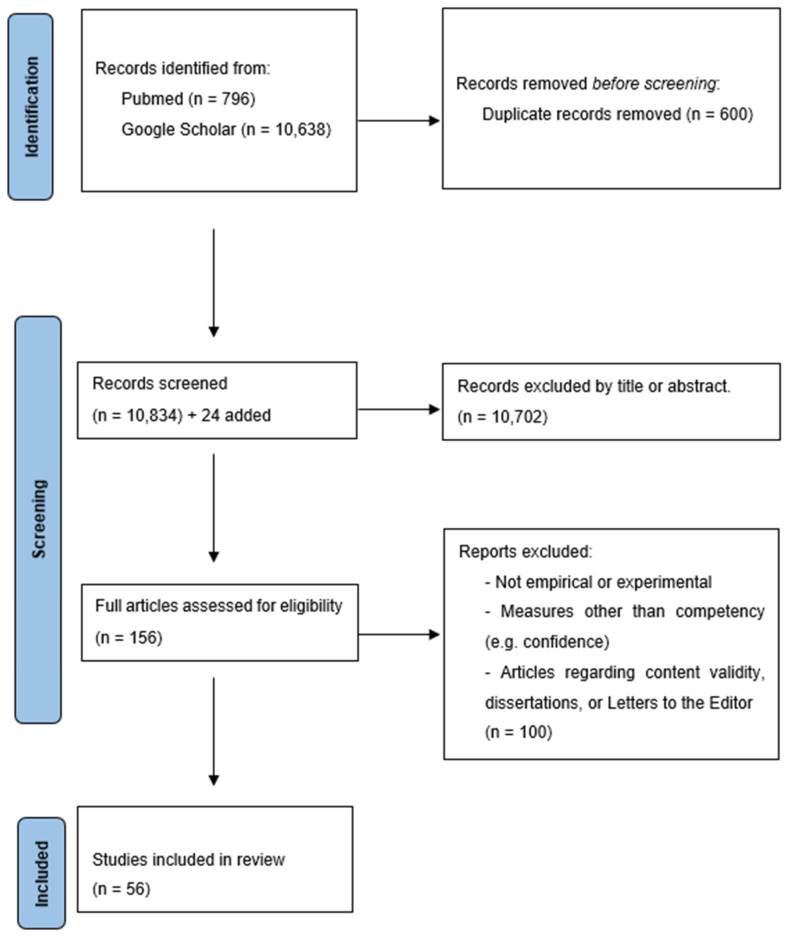
PRISMA Flowchart.

**Figure 2 behavsci-15-00468-f002:**
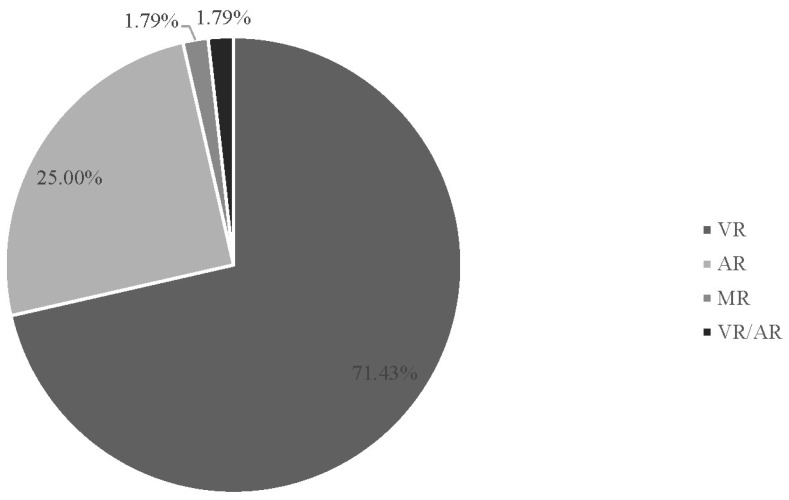
Percentage of immersive techniques used (l-XR).

**Figure 3 behavsci-15-00468-f003:**
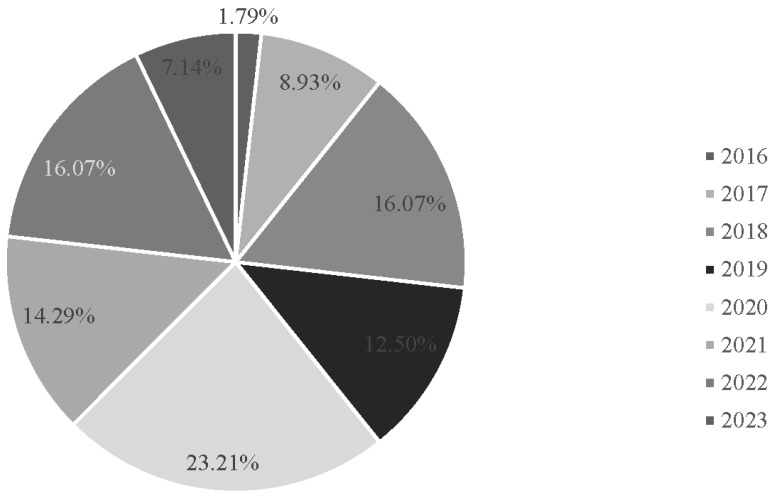
Percentage of studies by year.

**Figure 4 behavsci-15-00468-f004:**
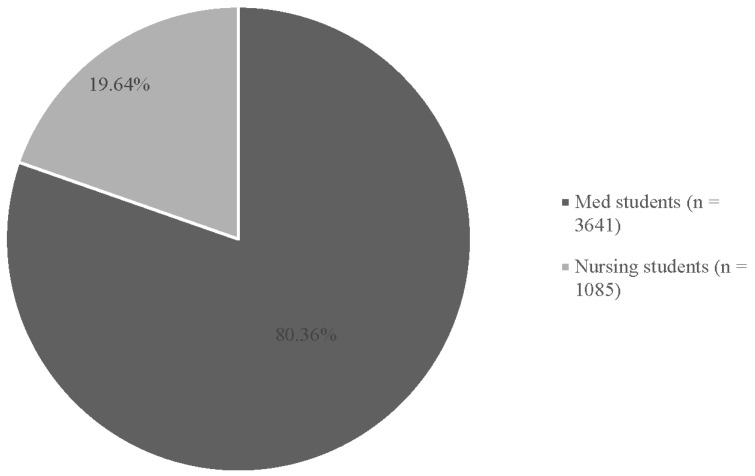
Percentage of student types reviewed.

**Figure 5 behavsci-15-00468-f005:**
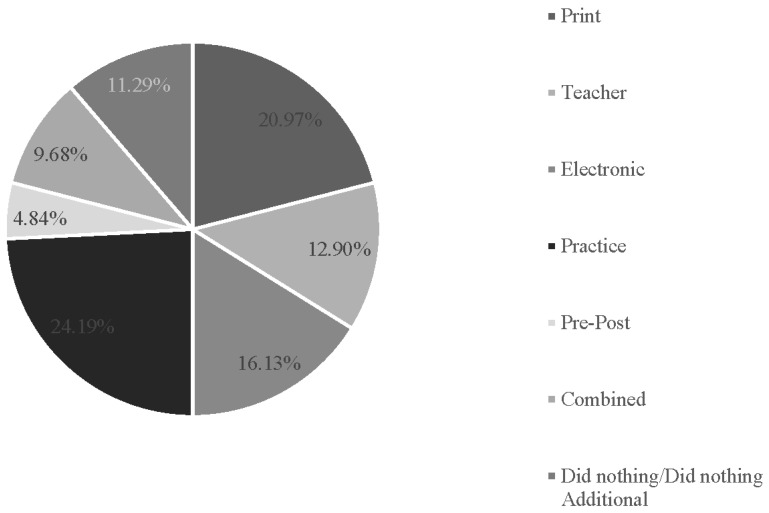
Percentage of types of control groups (n = 62 control groups across 56 studies).

**Figure 6 behavsci-15-00468-f006:**
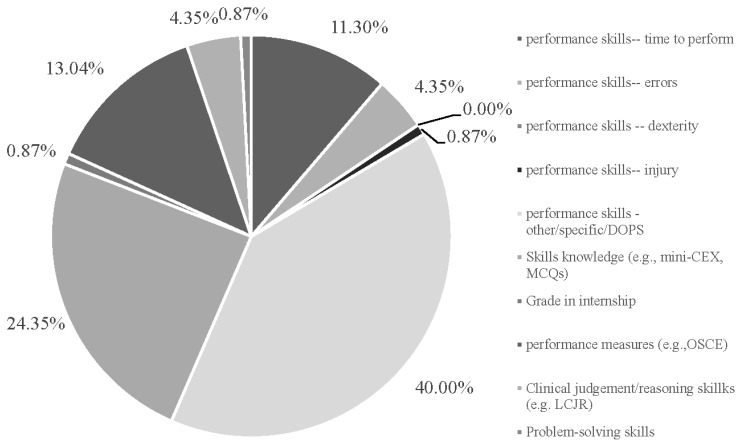
Percentage of outcomes (115 of 56 studies).

**Figure 7 behavsci-15-00468-f007:**
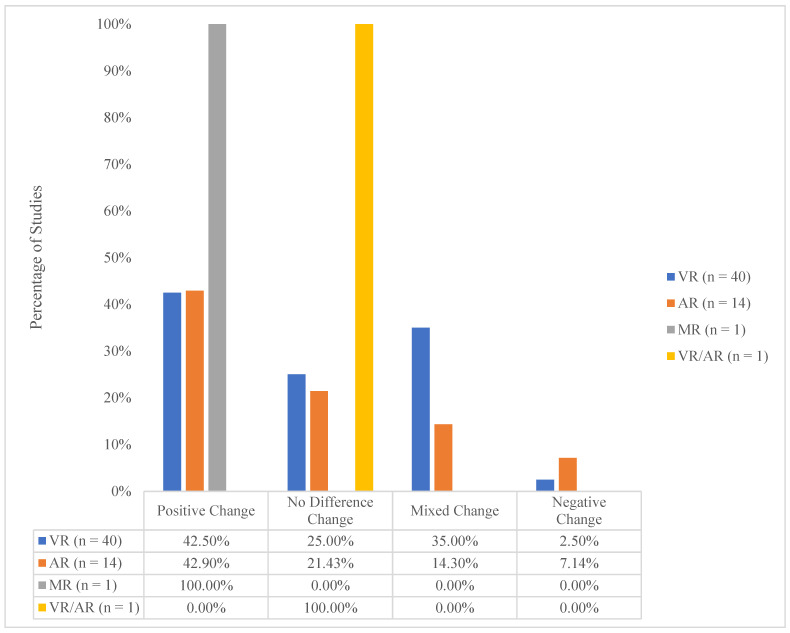
Percentage of VR/AR/MR Studies by Outcome Success of Clinical Skills. *Note*: “Positive Change” = total support for I-XR methods (vs. control) on all variables and outcomes, No Difference Change” = I-XR and control methods produced no differences on all variables and outcomes, “Mixed Change” = includes “Mixed Evidence Positive” (more support for I-XR outcomes than control methods on variables and outcomes) and “Mixed Evidence Negative” (more support for control methods than I-XR methods on variables and outcomes), “Negative Change” = total support for control methods (vs. I-XR) for all variables and outcomes.

**Table 1 behavsci-15-00468-t001:** Empirical Articles (56) Included in the Systematic Review.

Authors	Location	Type of I-XR	Population Studied/Setting	Number of Participants	Control Group	Basic Experimental Design/Description	Results	DVs	Outcome Evidence for I-XR	Study Evidence for I-XR
1. [3] ([3])	United States	AR	nursing − tube placement	69 nursing students	1 (Combined)	Nursing students were tested on their ability to place a nasogastric tube. They were randomly assigned to either usual training (which included both video and didactic content) or an iPad anatomy-augmented virtual simulation training module.	The AR group was able to more accurately and successfully place the NGT, *p* = 0.011. CONTROL: *n* = 34, M = 15.39 (SD = 1.01). EXPERIMENAL: *n* = 35, M = 15.96 (SD = 0.75).	1 (performance skills − specific)	+ Performance skills	Positive
2. [5] ([5])	Denmark	VR + Electronic *	Medical − catheter placement	19 medical students	1 (Did nothing additional)	Students were split into two different training groups: immersive virtual reality versus the control group. Both groups viewed videos showing ultrasound-guided peripheral venous cannulation placement. The control group was given no further training.	The immersive VR group was significantly more successful at peripheral venous cannulation placement in comparison to the control group, *p* ≤ 0.001. CONTROL: *n* = 9, M = 22.2% placement [0.11, 0.41]. EXPERIMENTAL: *n* = 10, M = 73.3% placement [0.56, 0.86].	1 (performance skills − specific)	+ Performance skills	Positive
3. [6] ([6])	Denmark	VR	medical − Ultrasound skills	104 medical students	1 (Teacher)	Medical students were divided into two groups to learn Point-of-care ultrasound (POCUS) skills: a self-directed immersive virtual reality (IVR) group versus an instructor-led learning group. US skills were then assessed according to an OSAUS test.	There were no significant differences between the self-directed IVR and instructor-led groups in terms of OSAUS scoring or any other subgroup objectives. Overall effect, *p* = 0.36. EXPERIMENTAL: *n* = 51, M = 10.3 [9.0, 11.5]. CONTROL: *n* = 53, M = 11.0 [9.8, 12.2].	1 (performance measures − OSCE)	X No difference in performance measures	No Difference
4. [8] ([8])	Netherlands	VR	medical − Obstetrics training	89 medical students	1 (Print)	Two weeks prior to medical students’ OB/Gyn internship, students underwent teaching on gentle Casarean Sections (gash) and general obstetric knowledge. Students were divided into either a control group that underwent conventional study or an experimental group that watched 360-degree videos using VR. After the internship, the authors analyzed the grades received for the internship, as well as administered both open-ended and multiple-choice question tests.	No significant difference in internship grade between groups, *p* = 0.66 (adjusted 0.68). EXPERIMENTAL: *n* = 53, M = 7.75 CONTROL: *n* = 48, M = 7.83 Mean difference CI [−0.33, 0.16]. No significant difference in multiple-choice testing (skills knowledge) between the groups, *p* = 0.91 (adjusted 0.68). EXPERIMENTAL: *n* = 53, M = 6.63. CONTROL: *n* = 48, M = 6.67 Mean difference CI [−0.61, 0.55].	2 (Grade internship) (Skills knowledge based on MCQs)	X No difference Grade internship X No difference Skills knowledge	No Difference
5. [10] ([10])	United States	AR	nursing − IV placement & Chest compression	20 nursing students	1 (Practice)	Students underwent either standard training or training with AR via a head-mounted display for learning needle chest decompression and IV-line placement skills. The students’ skills were measured with a post-assessment both immediately after training and 3 weeks later.	Results are assessed with respect to control immediately after. The AR head-mounted group displayed better needle chest decompression skills. No *p* value or M, SD reported. No significant difference in IV placement performance was found between groups. No *p* value or M or SD was reported. EXPERIMENTAL: *n* = 10 CONTROL: *n* = 10	2 (performance skills − specific chest) (performance skills- other IV)	+ Performance skills X No difference in performance skills	Mixed Positive
6. [11] ([11])	Canada	VR	Medical − surgery	40 medical students	2 (Practice) (Did nothing)	Medical students underwent five weeks of independent training sessions in one of three groups: a high-fidelity virtual reality arthroscopic simulator, a bench-top arthroscopic simulator, or an untrained group (control). To measure post-test skill acquisition, students performed a diagnostic arthroscopy on both simulators and were tested in a simulated intraoperative environment using a cadaveric knee. A more difficult surprise skills transfer test was also administered. Students were evaluated using the Global Rating Scale (GRS) and a timer to determine efficiency.	Results are not reported for crossover group post- training. Both the high-fidelity VR simulator and bench-top arthroscopic simulator groups showed significant improvement in arthroscopic skills compared to the control, *p* < 0.05 for both. The VR simulation group showed the greatest improvement in performance in the diagnostic arthroscopy crossover tests using the GRS), *p* < 0.001. CONTROL: *n* = not reported, M_D_ = 0.75 (SD only reflected in error bars). EXPERIMENTAL VR: *n* = not reported, M_D_ = 12.6 (SD only reflected in error bars). VR group showed the fastest improvement in simulated cadaveric setup with timer, *p* < 0.001. CONTROL: *n* = not reported, M_D_ = 9.1 (SD only reflected in error bars). EXPERIMENTAL VR: *n* = not reported, M_D_ = 17.3 (SD only reflected in error bars).	2 (performance measures GRS) (performance skills-time)	+ Performance measures + Performance skills	Positive
7. [15] ([15])	United States	VR	nursing − tracheostomy care	172 nursing students	1 (Combined)	Nursing students were divided into control and VR groups for tracheostomy care and skill knowledge. Both groups completed a theoretical class, labs, and small group study. The experimental group was provided with a game-based virtual reality phone application. Skills knowledge was assessed using the FEMA IS-346 exam, and performance skills were assessed using the Decontamination Checklist for performance.	Results for the less immersive VR are not reported. Only the immersive experimental group is compared to the control group. Both groups increased their skills performance after training but did not differ from one another, *p* = 0.443. CONTROL: *n* = 58, M = 13.48 (SD = 0.30). EXPERIMENTAL: *n* = 59, M = 14.24 (SD = 0.29). Both groups increased their skills and knowledge after training but did not differ from one another, *p* = 1.00. CONTROL: *n* = 58, M = 16.07 (SD = 0.30). EXPERIMENTAL: *n* = 59, M = 16.25 (SD = 0.29).	2 (performance skills − specific suctioning) (skills knowledge)	X No difference performance skills X No difference skills knowledge	No Difference
8. [18] ([18])	United States	VR	medical − surgery	20 medical students	1 (Print)	Medical students were randomized into either standard guide (SG) or virtual reality (VR) learning groups to learn intramedullary nailing (IMN) of the tibia. Students then performed a simulated tibia IMN procedure immediately following their training and were evaluated by an attending surgeon using a procedure-specific checklist and a 5-point global assessment scale. Students returned 2 weeks later for repeat training and testing.	The VR groups showed significantly higher global assessment scores, *p* < 0.001. CONTROL: *n* = 10, M = 7.5, SD = not reported. EXPERIMENTAL: *n* = 10, M = 17.5, SD = not reported. The VR also completed a higher percentage of steps correctly according to the procedure-specific checklist, *p* < 0.002. CONTROL: *n* = 10, M = 25, SD = not reported. EXPERIMENTAL: *n* = 10, M = 63, SD = not reported.	2 (performance skills − specific) (performance measures)	+ Performance skills + Performance measures	Positive
9. [19] ([19])	Netherlands	AR	BIO medical students − anatomy	58 (bio)medical students	2 (Print) (Practice)	Students were divided into three groups: (1) the stereoscopic 3D augmented-reality (AR) group, (2) the monoscopic 3D desktop model group, or (3) the 2D anatomical atlas group. Students were told what the learning goals consisted of and were given instructions for the session. Visual-spatial abilities were measured before the learning session began. Post-session learning was measured using a 30-question knowledge test that tested the factual, functional, and spatial organization of anatomical structures.	All groups performed equally well on the knowledge test, *p* = 1.00. Results are between the AR and the atlas control. CONTROL: *n* = 18, M = 50.9 (SD = 13.8). EXPERIMENTAL: *n* = 20, M = 47.8 (SD = 9.8).	1 (skills knowledge)	X No difference in skills knowledge	No Difference
10. [20] ([20])	Germany	AR	medical − anatomy	749 medical students	2 (Print) (Practice)	Medical students were divided into one of three groups: (1) the control group using radiology atlases, (2) a virtual dissection table, or (3) AR Magic Mirror. A pre and post-test was taken about anatomy questions.	Pre-post not evaluated for final assessment. Both the AR Magic Mirror group and the Theory (control) group showed significantly increased post-test scores but did not differ from one another. No *p* value for comparison between change in improvement was given. Results are from the post scores between the AR and the theory from the print control group. CONTROL: *n* = 24, M = 50.60 (SD = 12.53). EXPERIMENTAL: *n* = 24, M = 48.00 (SD = 13.07).	1 (skills knowledge)	X No difference in skills knowledge	No Difference
11. [21] ([21])	Germany	VR	med medical − surgery	36 medical students	1 (Practice)	Medical students underwent a 5-day laparoscopic basic skills training course using either a box trainer or virtual reality (VR) training curriculum. Skills were measured by students’ performance of an ex-situ laparoscopic cholecystectomy on a pig liver using RT and errors. The performance was evaluated by the Global Operative Assessment of Laparoscopic Skills (GOALS) score.	Both groups showed significant improvement in their acquisition of laparoscopic basic skills, and the two groups did not differ in improvement on the peg transfer, *p* = 0.311. CONTROL: *n* = 18, M = 53 (SD = 21.3). EXPERIMENTAL: *n* = 18, M = 44.4 (SD = 14.9). The two groups also did not differ in their pattern cutting, *p* = 0.088. CONTROL: *n* = 18, M = 31.6 (SD = 17.3). EXPERIMENTAL: *n* = 18, M = 42.6 (SD = 16.9). The two groups did not differ on loop placement, *p* = 0.174. CONTROL: *n* = 18, M = 46.3 (SD = 54). EXPERIMENTAL: *n* = 18, M = 53.1 (SD = 32.5). The two groups did not differ in their knot tying, *p* = 0.174. CONTROL: *n* = 18, M = 37.2 (SD = 11.9). EXPERIMENTAL: *n* = 18, M = 42.6 (SD = 16.4). The GOALS scores on four of the five items were significantly higher in the box-trained group compared to the VR-trained group (individual comparisons in Table 5 of the original publication).	5 (performance skills − other peg) (performance skills − other cutting) (performance skills − other loop) (performance skills − other knot) (performance measures- GOALS)	X No difference in performance skills X No difference in performance skills X No difference in performance skills X No difference in performance skills − Performance measures	Mixed Negative
12. [22] ([22])	Denmark	VR	medical- cystoscopy	32 medical students	1 (Teacher)	Two groups of medical students completed endoscopic procedure training. The control group underwent traditional lecture-based training, whereas the experimental group used VR and other self-directed simulation training methods. Three weeks after the training, participants performed cystoscopies on two patients, and performance was measured using a Global Rating Scale (GRS).	No significant difference in performance between the two groups was found after training, *p* = 0.63. CONTROL: *n* = 12, M = 14.3. EXPERIMENTAL: *n* = 13, M = 13.6. CI of the difference only reported: [−2.4, 3.9].	1 (performance measures − GRS)	X No difference in performance measures	No Difference
13. [23] ([23])	United States	VR	nursing − catheter	20 nursing students	1 (Combined)	Nursing students were assigned to either a control group (traditional learning with a task trainer) or an experimental group (VR software/game) to learn catheter insertion skills. Skills were assessed approximately two weeks after completion of the training session.	VR group completed more procedures than the traditional group, *p* < 0.001. CONTROL: *n* = 10, M = 1.8 (SD = 0.42). EXPERIMENTAL: *n* = 10, M = 3.0 (SD = 1.3). Pass rates at two weeks were identical; no *p* value was given.	2 (performance skills − other number of procedures completed) (performance skills − other specific pass rates)	+ Performance skills X No difference in performance skills	Mixed Positive
14. [24] ([24])	United States	VR	medical − surgery	20 medical students and orthopedic residents	1 (Combined)	Medical students and orthopedic residents were randomized into either standard guide (SG) or virtual reality (VR) learning groups to learn pinning of a slipped capital femoral epiphysis (SCFE), a pediatric orthopedic surgery procedure. All participants watched a technique video, and the VR group completed additional training on the Osso VR surgical trainer. Participants were then asked to achieve “ideal placement”, and performed a SCFE guidewire placement on Sawbones model 1161. Evaluation was based on time, number of pins “in-and-outs”, articular surface penetration, the angle between the pin and physis, distance from pin tip to the subchondral bone, and distance from the center-center point of the epiphysis.	The VR group showed superiority across multiple domains but were not statistically different from the control in the following: time to final pin placement, *p* = 0.26. CONTROL: *n* = 10, M = 706 (SD shown in Figure 1 in the original publication). EXPERIMENTAL: *n* = 10, M = 573 (SD shown in Figure 1 in the original publication). VR performed better compared to the control for pin in and out *p* = 0.26. CONTROL: *n* = 10, M = 1.7 (SD shown in Figure 1 in the original publication). EXPERIMENTAL: *n* = 10, M = 0.5 (SD shown in Figure 1 in the original publication). VR group performed fewer surface penetrations, *p* = 0.36. CONTROL: *n* = 10, M = 0.2 (SD shown in Figure 1 in the original publication). EXPERIMENTAL: *n* = 10, M = 0.4 (SD shown in Figure 1 in the original publication). VR group had a smaller distance pin to tip to subchondral bone, *p* = 0.49. CONTROL: *n* = 10, M = 5.8 (SD = 3.36). EXPERIMENTAL: *n* = 10, M = 7.2 (SD = 6.5). VR group had a lower angle deviation between the pin and physis, *p* < 0.05. CONTROL: *n* = 10, M = 4.9 (SD = 3.0). EXPERIMENTAL: *n* = 10, M = 2.5 (SD = 1.42).	5 (performance skills − time) (performance skills − other specific pin in and outs) (performance skills − errors) (performance skills − other specific pin tips to bone) (performance skills − other specific angle)	X No difference in performance skills X No difference in performance skills X No difference in performance skills X No difference in performance skills + Performance skills	Mixed Positive
15. [25] ([25])	Taiwan	VR	nursing − tube placement	45 nursing students	1 (Electronic)	Nursing students were randomly assigned into two groups to learn nasogastric (NG) tube feeding: (1) an immersive 3D interactive video program group or (2) a regular demonstration video. Students completed a pre- and post-intervention questionnaire, which included a nasogastric tube feeding quiz (NGFQ) to study NG tube feeding knowledge. Students were assessed after intervention and 1 mo. Later.	Knowledge scores on NG tube feeding improved significantly in both groups; however, there was no significant difference in the knowledge scores after treatment, *p* = 0.77 CONTROL: *n* = 23, M = 11.7 (SD = 1.86). EXPERIMENTAL: *n* = 22, M = 11.9 (SD = 2.04).	1 (skills knowledge)	X No difference in skill knowledge	No Difference
16. [26] ([26])	Taiwan	VR	Medical − intake skills	64 medical students	1 (Electronic)	Students were randomized into two groups and received either a 10 min immersive 360-degree virtual reality or a 2D virtual reality instructional video on history taking and physical examination skills. Within 60 min of watching the video, students performed a focused history and physical on a patient. The Direct Observation of Procedural Skills (DOPS) was used to measure physical exam skills, and the Mini-CEX was used to measure general history and physical exam skills.	The average DOPS-total score was significantly higher in the VR video group compared to the 2D video group, *p* = 0.01. CONTROL: *n* = 32, M = 85.8 (SD = 3.2). EXPERIMENTAL: *n* = 32, M = 88.4 (SD = 4.0). No significant differences in the average Mini-CEX scores were found between the groups, *p* = 0.75. CONTROL: *n* = 32, M = 39.8 (SD = 5.2). EXPERIMENTAL: *n* = 32, M = 40.1 (SD = 4.1).	2 (performance skills − other/DOPS) (skills knowledge − Mini-CEX)	+ Performance skills X No difference in skills knowledge	Mixed Positive
17. [29] ([29])	Taiwan	AR	Nursing students − first aid	95 nursing students	1 (Practice)	Nursing students were divided into two groups for pediatric first-aid training. The control group performed a simulation using a traditional Resusci Annie, whereas the experimental group used an interactive Resusci Anne that overlaid AR. Pre and post-tests were given to evaluate participant knowledge and skills. Knowledge was assessed using a 20-question test. Skill level was assessed using a graded evaluation checklist.	The AR intervention group showed significantly higher post-test knowledge, *p* < 0.001. CONTROL: *n* = 49, M = 18.08 (SD = 1.6). EXPERIMENTAL: *n* = 46, M = 18.78 (SD = 1.1). The AR group also showed improved skill in first-aid level scoring compared to the control group post-test, *p* < 0.001. CONTROL: *n* = 49, M = 29.71 (SD = 1.5). EXPERIMENTAL: *n* = 46, M = 32.52 (SD = 1.3).	2 (skills knowledge) (performance measures − other specific first aid)	+ Skills knowledge + Performance measures	Positive
18. [34] ([34])	Canada	VR + Print *	medical − neuroanatomy	64 medical students	1 (did nothing additional)	Medical students neuroanatomy learning with VR. Pre and post-intervention tests were given, including a post-test immediately after the study completion and one 5–9 days later.	Both groups showed significant improvement between pre- and post-test scores but no significant differences on the neuroanatomy test between the groups on either of the post-test results, *p* = 0.5. Means and SDs are not reported: T-statistic reported for control (*n* = 33) vs. VR (*n* = 31) post-training, *t*(62) = − 0.38.	1 (skills knowledge)	X No difference in skills knowledge	No Difference
19. [36] ([36])	United States	VR	medical − suturing	14 medical students	1 (Practice)	Students were assigned to one of two training groups: (1) the VBLaST-SS (virtual simulator) training group or (2) the FLS training group. Students then watched a video that taught the intracorporal suturing task they were going to be practicing. Students then performed the task on both systems to measure baseline performance. Students then practiced once a day, five days a week, for three weeks. Performance scoring was based on the original FLS scoring system.	Both training modalities showed significant performance improvement, but there were no significant differences in the group x time interaction, *p* = 0.20. Learning curves for both learning modalities were also similar. Means and SD are only shown in Figure 3 in original publication.	1 (performance skills − specific FLS)	X No difference in performance skills	No Difference
20. [38] ([38])	United States	VR	Nursing − case evaluation for COPD	81 nursing students	1 (Practice)	This study placed students in two groups: those using mannequin-based simulations and those using VR simulations. Participants completed a standardized patient encounter of a complex case involving a patient with COPD. Pre and post-intervention knowledge assessments were also performed using the LCJR and the C-SEI.	Students in both groups showed significant improvement in post-test knowledge assessment. Scores between the groups were not significantly different in the post-test knowledge assessment, *p* = 0.48. CONTROL: *n* = 14, M = 79.82 (SD = 17.63). EXPERIMENTAL: *n* = 14, M = 82.16 (SD = 11.76). There was no statistical difference post-intervention for either group for the LCJR, *p* = 0.374. CONTROL: *n* = 14, M = 82.69 (SD = 13.65). EXPERIMENTAL: *n* = 14, M = 78.18 (SD = 12.71). There was also no statistical difference post-intervention on the C-SEI between groups. CONTROL: *n* = 14, M = 84.62 (SD = 14.91). EXPERIMENTAL: *n* = 14, M = 81.93 (SD = 16.41).	3 (skills knowledge) (clinical reasoning − LCJR) (clinical reasoning − C-SEI)	X No difference in skills knowledge X No difference in clinical reasoning X No difference in clinical reasoning	No Difference
21. [39] ([39])	Republic of Korea	VR + SP	medical − neurological	95 medical students	1 (did nothing additional)	Medical students were divided into two groups: a standardized patient (SP) group that was provided neurological findings using conventional methods (verbal description, pictures, videos) versus an SP with Virtual Reality-based Neurological Examination Teaching Tool (VRNET) group. A researcher measured student performance using the Neurologic Physical Exam (NPE) score.	The SP + VR group had significantly higher NPE scores compared to the SP group, *p* = 0.043. CONTROL: *n* = 39, M = 3.40 (SD = 1.01). EXPERIMENTAL *n* = 59, M = 3.81 (SD = 0.92).	1 (skills knowledge − NPE score)	+ Skills knowledge	Positive
22. [41] ([41])	Netherlands	AR	medical and Biomedical- neuroanatomy	31 medical and biomedical students	1 (Print)	Students were assigned to one of two groups for learning neuroanatomy. The control group underwent learning with cross-sections of the brain, whereas the experimental group underwent AR learning.	Results are assessed with respect to control. The control group showed improved post-test scoring compared to the AR, *p* = 0.035. Results for adapted test scores: CONTROL: *n* = 16, M = 60.6 (SD = 12.4). EXPERIMENTAL: *n* = 15, M = 50.0 (SD = 10.2).	1 (skills knowledge)	− Skills knowledge	Negative
23. [44] ([44])	Taiwan	VR + workshop in ultrasound *	medical − Ultrasound skills	101 medical students	1 (Print)	Medical students took place in an ultrasonography (US) training program. They were divided into either the virtual reality (VR) intervention group or the control group. Both groups participated in an ultrasound workshop; however, the intervention group used a self-directed VR-enhanced anatomy review and used VR to complete additional review sessions during the US hands-on practice. After the US workshop was completed, participant competency was measured using a standardized practical US test, which focused on the identification of various anatomical structures, and a 10-Q MCQ on anatomy.	Participants in the intervention group showed significantly higher scores on US task performance overall, *p* < 0.01. The results below are for mean rank. No variability was given. CONTROL: *n* = 54, M_R_ = 38.52. EXPERIMENTAL: *n* = 47, M_R_ = 65.34. The VR group also showed significantly better scores on the knowledge test, *p* < 0.05. CONTROL: *n* = 54, median = 2 (IQR = 3). EXPERIMENTAL: *n* = 47, Median = 3 (IQR = 3).	2 (skills knowledge) (performance skills − other practical US)	+ Skills knowledge + Performance skills	Positive
24. [50] ([50])	Germany	VR	medical − CPR	160 medical students	1 (Practice)	Medical students were randomized into an intervention or control group. The intervention group completed a BLS course in virtual reality, whereas the control group underwent standard BLS training. At the end of training, all students performed a 3 min practical test using the Leardal Mannequin to record no flow time on the task.	The control group had significantly shorter no-flow time compared to the VR, *p* < 0.0001. CONTROL: *n* = 104, M = 82.03 (SD = not reported). EXPERIMENTAL: *n* = 56, M = 92.96 (SD = not reported).	1 (performance skills − specific, no flow time)	- Performance skills	Negative
25. [53] ([53])	Poland	VR + Practice *	medical − CPR	91 medical students	1 (Teacher)	Both the control and experimental groups completed a 3-h BLS course including background training and practice on a CPR mannequin. Students then participated in either a traditional teaching or VR scenario where hands-only CPR was completed. The quality of the chest compressions (rate and depth) was then tested and analyzed.	There were no significant differences in chest rate compression performance between the control and virtual reality groups, *p* = 0.48. CONTROL: *n* = 45, Median = 114 (IQR 108–122). EXPERIMENTAL: *n* = 45, Median = 115 (IQR 108–122). There was also no significant difference in chest rate depth between groups, *p* > 0.05. CONTROL: *n* = 45, Median = 48 (IQR = 44–55). EXPERIMENAL: *n* = 45, Median = 49 (IQR = 43–53). Finally, there was a significant increase in the percentage of chest compression relaxation for the control group compared to the VR group, *p* < 0.01. CONTROL: *n* = 45, Median = 97 (IQR = 85–100). EXPERIMENTAL: *n* = 45, Median = 69 (IQR = 26–98).	3 (performance skills − specific rate) (performance skills − specific depth) (performance skills − specific relaxation)	X No difference in performance skills X No difference in performance skills - Performance skills	Mixed Negative
26. [59] ([59])	Republic of Korea	VR	Nursing students − chemport insertion surgery	60 nursing students	1 (Print)	Nursing students were divided into two groups to learn chemoport insertion surgery. The control group’s learning consisted of instruction by an operating nursing instructor, learning via a handout, and time for self-study. The experimental group used VR. Pre and post-test knowledge was assessed using a 10-point questionnaire about key knowledge of insertion.	The VR group showed significantly higher post-test knowledge scores compared to the control group after training, *p* = 0.001. CONTROL: *n* = 30, M = 4.80 (SD = 1.65). EXPERIMETNAL: *n* = 30, M = 6.97 (SD = 1.35).	1 (skills knowledge)	+ Skills knowledge	Positive
27. [60] ([60])	Ireland	VR	med students − obstetrics	69 medical students	1 (Electronic)	Medical students were placed into one of two groups to help them learn and conceptualize fetal lies and presentation. The interventional group was immersed in a virtual reality learning environment (VRLE) to explore fetal lie, and the control group used traditional 2D images. After their sessions, clinical exam skills were tested using an obstetric abdominal model. Knowledge was assessed by students’ ability to determine fetal lies and presentation on this model. Time taken to complete the test was also measured.	No significant differences were found between the two groups in terms of knowledge assessment, although the authors note that there was a noticeable trend of higher success rates in the intervention group in the VR group compared to the control group) for a combined lie and presentation scores, *p* = not reported. CONTROL: *n* = 34, M = 70.0 (SD = not reported). EXPERIMENTAL: *n* = 33), M = 56 (SD = not reported). However, the time to complete the task was significantly less in the intervention group compared to the control group, *p* = 0.012. CONTROL: *n* = 34, M = 38 (SD = 10.83). EXPERIMENTAL: *n* = 33, M = 45 (SD = 12.95).	2 (performance skills − time) (performance skills -specific success)	X No difference in performance skills + Performance skills	Mixed Positive
28. [68] ([68])	Germany	VR	medical − surgery	100 medical students	1 (Did nothing)	Medical students were divided into three groups to complete laparoscopic training: (1) the control group, which received no training; (2) the “alone” group, and (3) the dyad group. Intervention groups completed box and VR training, after which performance was measured with a cadaveric porcine laparoscopic cholecystectomy (LC), and the objective structured assessment of technical skills (OSATS) was used. Global operative assessment of laparoscopic skills (GOALS), time to complete LC, and VR performances were also measured.	Results are reported for improvement between the VR and control groups only. The VR group and the control group did not differ on the OSATS, *p* = 0.548. CONTROL: *n* = 20, M = 37.1 (SD = 7.4). EXPERIMENTAL: *n* = 40, M = 40.2 (SD = 9.8). The two groups did not differ on the GOALS either, *p* = 0.998. CONTROL: *n* = 20, M = 10.1 (SD = 3.0). EXPERIMENTAL: *n* = 40, M = 10.6 (SD = 3.0). The VR groups were faster than the control group in completion time, *p* < 0.001. CONTROL: *n* = 20, M = 13.5 [11.8, 17.5]. EXPERIMENTAL: *n* = 40, M = 10.2 [7.9, 11.3]. The VR group also had fewer movements, *p* = 0.002. CONTROL: *n* = 20, M = 871 [637, 1105]. EXPERIMENTAL: *n* = 40, M = 683 [468, 898]. The VR group also had a shorter path length, *p* = 0.004. CONTROL: *n* = 20, M = 1640 [1174, 2106]. EXPERIMENTAL: *n* = 40, M = 1316 [948, 1684].	5 (performance measures -OSATS) (performance measures − GOALS) (performance skills − time) (performance skills − other specific path) (performance skills − other specific length)	X No difference in performance measures X No difference in performance measures + Performance skills + Performance skills + Performance skills	Mixed Positive
29. [69] ([69])	Turkey	AR	medical − anatomy	70 medical students	1 (Print)	Medical students were placed into a control group, which used traditional teaching methods (textbook), or an experimental group, which used mobile augmented reality (mAR) technology (MagicBook) to learn neuroanatomy. Post-intervention knowledge was measured using an Academic Achievement Test (AAT), a 30-question multiple-choice test.	The experimental mAR group showed significantly superior performance on the AAT test, *p* < 0.05. CONTROL: *n* = 34, M = 68.34 (SD = 12.83). EXPERIMENTAL: *n* = 36, M = 78.14 (SD = 16.19).	1 (performance measures − AAT)	+ Performance measures	Positive
30. [70] ([70])	Singapore	VR	Nursing students	34 nursing students	1 (Nothing) (pre-post)	Nursing students participated in pre-test and post-test knowledge questionnaires regarding both subcutaneous insulin injection and intravenous therapy. After completing the pre-test, participants underwent learning using immersive VR before completing the post-test questionnaire.	There was a significant improvement in knowledge test scores (IV and subcutaneous injection) after training, *p* = 0.075. Only z-scores were reported with ranks. No M or SDs are reported pre vs. post.	1 (skills knowledge)	+ Skills knowledge	Positive
31. [73] ([73])	Canada	AR	medical − suturing	44 medical students	2 (Teacher) (Combined)	Students were randomized into one of three intervention groups to learn suturing skills: (1) faculty-led, (2) peer tutor-led, or (3) holography-augmented intervention arms (Suture Tutor). Outcomes measured include the number of simple interrupted sutures that were placed to achieve proficiency, the total number of full-length sutures used, and the time to achieve proficiency.	Results are reported among three groups. No significant differences among the intervention groups in the number of full-length sutures used, *p* = 0.376. Means and SD are reported for Teacher control vs. AR CONTROL: *N* = 16, M = 80.0 (SD = 47.2). EXPERIMENTAL: *n* = 14, M = 107.4 (SD = 61.3). No significant differences among the intervention groups in the number of simple interrupted sutures placed, *p* = 0.735. CONTROL: *n* = 16, M = 9.3 (SD = 5.3). EXPERIMETNAL: *n* = 14, M = 11.0 (SD = 6.0). No significant differences among the intervention groups in the time to achieve proficiency, *p* = 0.390. CONTROL: *n* = 16, M = 158.1 (SD = 89.2). EXPERIMETNAL: *n* = 14, M = 205.0 (SD = 113.2).	3 (performance skills − other specific number placed) (performance skills − other specific interruptions) (performance skills − other specific # used)	X No difference in performance skills X No difference in performance skills X No difference in performance skills	No Difference
32. [76] ([76])	United Kingdom	AR	Medical − surgery	24 medical students	1 (Practice)	This study simulated total hip arthroplasty (THA) and placed students in one of two groups to determine what training is more effective at improving the accuracy of acetabular component positioning: (1) augmented reality (AR) training (with live holographic orientation feedback) or (2) hands-on training with a hip arthroplasty surgeon. Students participated in one baseline assessment, training session, and reassessment a week for four weeks and were recorded on the target angle (inclination − anteversion).	AR intervention group showed smaller average errors than the control practice with the surgeon group after training, *p* < 0.0001. CONTROL: *n* = 12, M = 6 (SD = 4). EXPERIMENTAL: *n* = 12, M = 1 (SD = 1). In the final session, both groups showed improvement on the target angle, but there was no significant difference in performance between groups, *p* = 0.281. Means and SD are reported as differences from pre. CONTROL: *n* = 12, M_D_ = −8.4 [−7.0, −9.8]. EXPERIMENTAL: *n* = 12, M_D_ = −7.8 [−5.5, −10.2].	2 (performance skills − errors) (performance skills − other specific target angles)	+ Performance skills X No difference in Performance skills	Mixed Positive
33. [82] ([82])	Canada	VR immersive + VR simulation*	medical − anatomy	42 medical students	1 (Print)	Before learning cardiac anatomy, medical students underwent a VR simulation of the subject. Students then were separated into either a control group that continued independent anatomy study or an experimental group that underwent an immersive VR experience. Pre and post-test scores were obtained, which measured both conventional and visual-spatial (VS) cardiac anatomy questions.	Students in the immersive VR intervention group scored significantly higher overall after the intervention, *p* < 0.001. Mean differences between the control (*n* = 14) and experimental group (*n* = 28) are only reported: M_D_ = 24.8 (SD = 3.89). This included both subsections of VS and conventional content.	1 (skills knowledge)	+ Skills knowledge	Positive
34. [85] ([85])	Germany	VR	medical − BLS	88 medical students	1 (Electronic)	Medical students were placed into one of two groups to compare a control group that received web-based basic life support (BLS) training to an intervention group that underwent additional individual virtual reality (VR) training. The quality of BLS skills was assessed after training with a no-flow-time indicator. Overall, BLS performance was also assessed using an adapted observational checklist, graded by experts.	The VR intervention group showed significantly lower no-flow-time compared to the control, *p* = 0.009. CONTROL: *n* = 42, M = 11.05 (SD = 10.765). EXPERIMENTAL: *n* = 46, M = 6.46 (SD = 3.49). The VR group also showed significantly superior overall (lower penalty point) for their BLS performance in comparison to the control group, *p* < 0.001. CONTROL: *n* = 42, M = 29.19 (SD = 16.31). EXPERIMENTAL: *n* = 46, M = 13.75 (SD = 9.66).	2 (performance skills − other specific flow time) (performance measures − checklist)	+ Performance skills + Performance measures	Positive
35. [86] ([86])	Australia	AR/VR	medical − anatomy	59 medical and health science students	1 (Electronic)	Medical students completed a lesson on skull anatomy using one of three learning modalities: (1) virtual reality (VR), (2) augmented reality (AR), or (3) tablet-based (TB). After their 10min anatomy lesson using their respective learning modality, students completed a 20-question multiple-choice anatomy test to measure their knowledge.	No significant differences in the anatomy test scores were found between the three groups; AR and VR intervention did not show superior knowledge acquisition in comparison to tablet-based skull anatomy learning after the intervention, *p* = 0.874. Means are for the VR group and control group only. Standard deviations are depicted only in Figure 1 in the original publication. CONTROL: *n* = 22, M = 61. EXPERIMETNAL: *n* = 20, M = 59.	1 (skills knowledge)	X No difference in skills knowledge	No Difference
36. [87] ([87])	Japan	AR	medical − surgery	38 medical students	1 (Electronic)	Medical students were randomized into one of two groups for self-training suturing learning: (1) the augmented reality (AR) training group or (2) the instructional video group. Both groups watched an instructional video on subcuticular interrupted suturing and took a pre-test. They then practiced the suture 10 times using their assigned learning modality before completing a post-test. Pre- and post-tests were performed on a skin pad and were graded using global rating and task-specific subscales.	Both groups showed significant improvement between pre-test and post-test scores in both global rating and task-specific subscales on suturing performance; however, no significant difference in performance was found between the AR and instruction video training groups using the global rating, *p* = 0.38. CONTROL: *n* = 19, M = 15.11 (SD = 2.84). EXPERIMENTAL: *n* = 19, M 15.03 (SD = 1.94).	1 (performance skills − other specific suturing performance)	+ Performance skills	Positive
37. [88] ([88])	Germany	VR	medical − surgery	51 medical students	1 (Electronic)	Medical students were randomized into two groups for cystoscopy (UC) and transurethral bladder tumor resection (TURBT) training. The control group watched video tutorials by an expert. After completion of the training, students performed a VR-UC and VR-TURBT performance task and 12 measures of performance were recorded.	Both groups improved on three variables after training, including significantly lower average procedure length, lower resectoscope movement, and accidental bladder injury, but there was only one significant difference in the improved performances for the VR compared to the control (procedure time, *p* = 0.04). All Means and SD are listed in Table 2 of the original publication.	12 (performance skills − other specific × 9) (performance skills − injury) (performance skills − time × 2)	X No difference in performance skills (for 11 of the 12 variables)	No Difference
38. [90] ([90])	Denmark	VR	medical − Ultrasound skills	20 medical students	1 (Electronic)	Medical students were randomized into either a virtual reality (VR) or e-learning group for ultrasound education and training. Performance was scored using the OSAUS.	The VR group showed significantly higher scoring on the OSAUS compared to the e-learning group after intervention, *p* ≤ 0.001. CONTROL: *n* = 9, M = 125.7 (SD = 16.2). EXPERIMENTAL: *n* = 11, M = 142.6 (SD = 11.8).	1 (performance measures − OSAUS)	+ Performance measures	Positive
39. [91] ([91])	Germany	AR	medical − dermatology	44 medical students	1 (Print)	Medical students were randomized into one of two groups for learning dermatological knowledge. Group A’s training involved the use of a mobile augmented reality (mAR) application, whereas Group B’s training involved textbook-based learning. Baseline and post-test knowledge were assessed using a 10-question single choice (SC) test, which was repeated after 14 days to assess longer-term retention.	The initial SC post-test showed significant knowledge gain in both groups, but the VR group showed a marginally significant memory for correct answers after two weeks, compared to the control group, *p* = 0.10. CONTROL: *n* = 22, M = 0.33 (SD = 1.62). EXPERIMENTAL: *n* = 22, M = 1.14 (SD = 1.30).	1 (skills knowledge)	+ (marginal) Skills knowledge	Positive
40. [92] ([92])	United States	VR+	Medical − surgery	25 medical students	1 (Print)	Medical students were randomized into one of three groups for learning intramedullary tibial nail insertion: (1) the technique guide-only control group, (2) the virtual reality (VR) only group, or (3) the VR plus technique guide group. The experimental groups participated in three separate VR simulations, 3–4 days apart. After 10–14 days of preparation, students performed an intramedullary tibial nail insertion simulation into a bone-model tibia. Completion and accuracy were assessed.	Overall assessments are made with comparisons to both VR groups compared to the control group. Both experimental groups that involved VR training showed higher completion rates, *p* = 0.01 Both groups also had fewer incorrect steps, *p* = 0.02, in comparison to the control group. The Means and SD below are for VR+ and the control group only. ERRORS CONTROL: *n* = 8, M = 5.7 (SD = 0.2). EXPERIMENTAL: *n* = 9, M = 3.1 (SD = 0.1). COMPLETION TIME CONTROL: *n* = 8, M = 24 (SD = 4). EXPERIMENTAL: *n* = 9, M = 18 (SD = 8).	2 (performance skills − errors) (performance skills − time)	+ Performance skills + Performance skills	Positive
41. [94] ([94])	Germany	VR ‘high’	Nursing students- endotracheal suctioning skills	131 nursing students	2 (low VR) + Electronic	Nursing students were split into one of three groups for learning of endotracheal suctioning skills. The control group’s intervention was a video tutorial. The intervention groups consisted of a VR low group and a VR high group. The VR ‘low’ group’s intervention consisted of basic VR technology, including head tracking and controller-based controls. The VR ‘high’ group’s intervention consisted of more advanced VR technology, including head and hand tracking, allowing users to interact with their actual hands, as well as supplementing with real-world video clips. Participants were assessed using a knowledge test and a skill demonstration test on a manikin using an objective structured clinical examination (OSCE). The knowledge test was given immediately after intervention and 3 weeks later.	Each of the three groups showed a significant increase in knowledge acquisition; however, there was no significant difference among them for skills knowledge, *p* = 0.730. Means and SDs are for the VR+ vs. control group after intervention. CONTROL: *n* = 43, M = 7.16 (SD = 2.29). EXPERIMENTAL: *n* = 47, M = 7.06 (SD = 1.42). There was a significant difference among groups on the OSCE after intervention, *p* < 0.001. Means and SDs are for the VR+ vs. control group after intervention. CONTROL: *n* = 43, M = 11.95 (SD = 1.65). EXPERIMENTAL: *n* = 47, M = 9.41 (SD = 2.70).	2 (skills knowledge) (performance measures – OSCE)	X No difference in skills knowledge − Performance measures	Mixed Negative
42. [96] ([96])	Spain	VR	nursing − triage skills	67 nursing students	1 (Practice)	This study compared the use of VR to clinical simulation to determine the efficiency in executing the START (Simple Triage and Rapid Treatment) triage.	There were no significant differences between the VR and clinical simulation groups in the percentage of victims that were correctly triaged, *p* = 0.612 CONTROL: *n* = 35, M = 88.3 (SD = 9.65). EXPERIMENTAL: *n* = 32, M = 87.2 (SD = 7.2).	1 (performance skills − specific triage variables)	X PNo difference in performance skills	No difference
43. [100] ([100])	France	VR + Print	medical − surgery	176 medical students	1 (Did nothing additional)	All students were given a technical note detailing an external ventricular drainage (neurosurgical) technique. Students were randomized into two groups, one of which received no additional training and one which used immersive virtual reality (VR) as supplemental teaching. Knowledge was assessed with a multiple-choice test immediately after training and six months later.	VR training showed significantly superior knowledge gain after both initial assessment and at the six-month mark, *p* = 0.01. Means and SD are reported after the initial training. CONTROL: *n* = 88, M = 4.59, (SD = 1.4). EXPERIMENTAL: *n* = 85, M = 5.17 (SD = 1.29).	1 (skills knowledge)	+ Skills knowledge	Positive
44. [101] ([101])	France	VR	Medical − lumbar puncture	89 medical students	1 (Teacher)	Medical students were randomized into one of two groups to complete training on how to perform a lumbar puncture. The control group participated in traditional lecture learning, whereas Group 2 participated in immersive VR 3D video filmed from a first-person point of view (IVRA-FPV). After training, students performed a simulated lumbar puncture on a mannequin to analyze their applied learning skillset. An oral examination was also included as an assessment.	The group that participated in the traditional lecture showed significantly superior scoring in the oral examination, *p* < 0.001. CONTROL: *n* = 44, M = 4.97, SE = 0.10. EXPERIMENTAL: *n* = 45, M = 4.06, SE = 0.12. The VR group took less time to perform the simulated lumbar puncture compared to the control, *p* < 0.01. CONTROL: *n* = 55, M = 73, SE = not reported. EXPERIMENTAL: *n* = 36, M = 50, SE = not reported. The VR group also had reduced errors compared to the control, *p* < 0.01. Means are reported as latency of errors. CONTROL: *n* = 43, M = 227.50, SE = 34.34. EXPERIMENTAL: *n* = 44, M = 153.26, SE = 11.19	3 (skills knowledge) (performance skills − time) (performance skills − errors)	− Skills knowledge + Performance skills + Performance skills	Mixed Positive
45. [103] ([103])	Germany	MR + Practice	medical − catheter placement	164 medical students	1 (Did nothing additional)	Medical students were randomized into one of two groups to undergo bladder catheter placement learning. One group underwent learning with an instructor, while the other group received mixed reality (MR) training using Microsoft HoloLens. Both groups were able to participate in hands-on training before undergoing a standardized objective structured clinical examination (OSCE) for performance assessment.	The MR intervention group showed significantly superior bladder catheter placement simulation in comparison to the control group, *p* = 0.000. CONTROL: *n* = 107, M = 19.96, (SD = 2.42). EXPERIMENTAL: *n* = 57, M = 21.49, (SD = 2.27)	1 (performance measures − OSCE)	+ Performance measures	Positive
46. [106] ([106])	China	VR	Medical − anatomy	30 medical students	1 (Combined)	Students were divided into either a traditional teaching group, or a virtual reality (VR) teaching group for teaching skull base tumors and skull anatomy. The traditional teaching group used literature-based learning, problem-based teaching, and case-based teaching, whereas the VR groups used real case images and Hololens (VR) glasses after the completion of their intervention.	VR group had a higher total scoring on the basic knowledge assessment compared to the traditional control group, *p* < 0.001. CONTROL: *n* = 15, M = 63.6 (SD = 3.81). EXPERIMENTAL: *n* = 15, M = 77.07 (SD = 4.00).	1 (skills knowledge)	+ Skills knowledge	Positive
47. [109] ([109])	Turkey	VR immersive or VR computer	nursing − decontamination skills	172 nursing students	1 (Print)	Nursing students were divided into three groups for decontamination skills training. The control group used a traditional written instructions learning method, whereas the experimental groups underwent immersive VR or computer-based VR training. Post-learning competency was measured using a Decontamination Checklist in which students performed skills on a mannequin. Cognitive test scores, performance scores, and time to complete skills were measured immediately post-training and 6 months later.	Results are reported for the immediate follow-up post-intervention. There were no significant differences among groups for the cognitive scores, *p* = 0.568. Means and SD are shown both VR groups combined and the traditional control group only. CONTROL: *n* = 43, M = 19 [8, 23]. EXPERIMENTAL: *n* = 43, M = 19 [13, 23]. There was no difference among the groups for time to completion on the OSCE, *p* = 0.723. Medians and ranges are shown in both VR groups combined and the traditional control group only. CONTROL: *n* = 43, Median = 260 (range: 190, 360). EXPERIMENTAL: *n* = 43, Median = 260 (range: 180, 360). The computer/mouse VR groups showed superior performance measures compared to the control group on the immediate post-test, *p* = 0.017. Medians and ranges are shown in both VR groups combined and the traditional control group only. CONTROL: *n* = 43, Median = 54 (range: 46, 57). EXPERIMENTAL: *n* = 43, Median = 55 (range: 46, 57).	3 (skills knowledge) (performance skills − time) (performance skills - other specific decontainment mannikin)	X No difference in skills knowledge X No difference in performance skills − Performance skills	Mixed Negative
48. [110] ([110])	United States	VR	medical − neuroanatomy	66 medical students	1 (Electronic)	Medical students were assigned to either a control (online textbook) or an experimental (3D imaging VR interactive model) group for the learning of neuroanatomy.	Students completed preintervention, post-intervention, and retention tests for assessment of knowledge. No significant differences in anatomy knowledge assessments were found between the control and VR groups, *p* = 0.87. CONTROL: *n* = 33, M = 0.76 (SD = 0.14) EXPERIMENTAL: *n* = 33, M = 75 (SD = 0.16)	1 (skills knowledge)	X No difference in skills knowledge	No difference
49. [112] ([112])	Saudi Arabia	VR	Medical − tube placement	169 medical students	1 (Practice)	Medical students were split into two groups to participate in a learning workshop regarding communication and collaboration. The workshop was a half-day, once a week, for 6 months. VR group received VR instruction, whereas the control group received conventional learning (simulated patients, lectures). Post-intervention assessment included an MCQs score and an Objective Structured Clinical Examinations (OSCE) score.	The VR intervention groups showed significantly higher MCQs compared to the control group after training, *p* < 0.001. CONTROL: *n* = 112, M = 15.9 (SD = 2.9). EXPERIMENTAL: *n* = 57, M = 17.4 (SD = 2.1). The VR group also showed improved OSCE after training compared to the conventional learning group, *p* < 0.001. CONTROL: *n* = 112, M = 9.8 (SD = 4.2). EXPERIMENTAL: *n* = 57, M = 12.9 (SD = 4.1).	2 (skills knowledge) (performance measures-OSCE)	+ Skills knowledge + Performance measures	Positive
50. [118] ([118])	Japan	VR	medical students − basic clinical knowledge	210 medical students	1 Nothing (pre-post)	Medical students participated in a lecture that used Virtual Patient Simulations (VPSs). Pre- and post-test 20-item multiple-choice questionnaires were taken and involved both knowledge and clinical reasoning items.	Students showed a significant increase in post-test scoring on both knowledges, *p* = 0.003. EXPERIMENTAL_pre_: *n* = 169, M = 4.78 [4.55, 5.01]. EXPERIMENTAL_post_: *n* = 169, M = 5.12 [4.90, 5.43]. Students also had increased clinical reasoning after training, *p* < 0.001. EXPERIMENTAL_pre_: *n* = 169, M = 5.3 [4.98, 5.58] EXPERIMENTAL_post_: *n* = 169, M = 7.81 [7.57, 8.05].	2 (skills knowledge) (clinical reasoning)	+ Skills knowledge + Clinical reasoning	Positive
51. [124] ([124])	Switzerland	AR	Medical − surgery	21 medical students	1 (Print)	Medical students were recruited for training in extracorporeal membrane oxygenation (ECMO) cannulation. They were split into two groups: (1) conventional training instructions for the first procedure and AR instructions for the second; (2) reverse order (AR instructions for the first procedure). Participants performed the two ECMO cannulation procedures on a simulator. Training times and a detailed error protocol were used for assessment.	AR group showed minimally higher training times compared to the control group, with no *p* values given. Means and SD in Figure 5 in original publication. AR group had significantly fewer errors when performing the second (more complex) simulation procedure, no *p* values given. Means and SD in Figure 5 in original publication.	2 (performance skills − error) (performance skills − time)	+ Performance skills - Performance skills	Mixed
52. [126] ([126])	Republic of Korea	VR	nursing − infant respiration	83 nursing students	2 (Teacher) (Practice)	Nursing students were separated into three groups to undergo neonatal resuscitation training. These groups included a virtual reality group, a high-fidelity simulation group, and a control (online lectures only) group. Pre and post-test scores were analyzed on neonatal resuscitation knowledge, problem-solving ability, and clinical reasoning ability.	Knowledge scores increased for all groups post-intervention, but the VR and simulation groups showed significantly higher knowledge after intervention compared to the control group, *p* = 0.004. Means and SDs are reported for the VR and control groups only. CONTROL: *n* = 26, M = 11.85 (SD = 5.43). EXPERIMENTAL: *n* = 29, M = 18.00 (SD = 2.55). The VR group showed significantly improved problem-solving ability scores in comparison to both the simulation and control groups, *p* = 0.038. CONTROL: *n* = 26, M = 106.24 (SD = 24.52). EXPERIMENTAL: *n* = 29, M = 122.72 (SD = 15.68). Clinical reasoning ability showed significantly improved performance, but none of the groups differed statistically, *p* = 0.123. CONTROL: *n* = 26, M = 53.69 (SD = 12.02). EXPERIMENTAL: *n* = 29, M = 59.66 (SD = 9.44).	3 (skills knowledge − other specific resuscitation) (problem-solving) (clinical reasoning)	+ Skills knowledge + Problem-solving X No difference in clinical reasoning	Mixed Positive
53. [127] ([127])	Canada	AR + Ultrasound	medical- lumbar puncture and facet joint injection	36 medical students	1 (Practice)	Medical students were randomized into either a control group (ultrasound use only), or an experimental group that involved training with ultrasound and AR using Perk Tutor software. The training involved learning lumbar puncture and facet joint injection skills on five different tasks for two tasks, a simpler and hard one (ultrasound-guided facet joint injection).	Results are reported for the harder task only. The AR group had more successful injections *p* = 0.04. CONTROL: *n* = 10, M = 37.5, SD = not reported. EXPERIMETNAL: *n* = 10, M = 62.5, SD = not reported. AR was also better than control post-training for total time, *p* < 0.001. CONTROL: *n* = 10, M = 103 (SD = 13). EXPERIMETNAL: *n* = 10, M = 47 (SD = 3). AR was also better than control post-training for time inside the phantom body, *p* < 0.01. CONTROL: *n* = 10, M = 31 (SD = 5). EXPERIMETNAL: *n* = 10, M = 14 (SD = 2). AR was also better than the control post-training for path distance inside the phantom body, *p* < 0.01. CONTROL: *n* = 10, M = 266 (SD = 76). EXPERIMETNAL: *n* = 182, M = 47 (SD = 36). AR was also better than control post-training for potential tissue damage, *p* = 0.03. CONTROL: *n* = 10, M = 3217 (SD = 1173). EXPERIMETNAL: *n* = 10, M = 2376 (SD = 673).	5 (performance skills − time) (performance skills − other specific time inside) (performance skills − other specific path) (performance skills − other specific damage) (performance skills − other specific success)	+ Performance skills + Performance skills + Performance skills + Performance skills + Performance skills	Positive
54. [128] ([128])	China	VR + Practice (BOX) *	Medical − surgery	51 medical students	1 Nothing (pre-post)	Medical students completed four pre and post-experiments with a box trainer-based laparoscopic surgery simulators (VRLS). Students were assessed by expert surgeons using the Global Operative Assessment of Laparoscopic Skills (GOALS) standards on performance and time on two tasks, the fundamental task (FT) and the color resection task (CRT)	Post-test assessments showed a significantly faster task completion post-training on both tasks, *p* < 0.01. Means and SDs are shown pre- and post-for-one- task (FT). EXPERIMENTAL_pre_: *n* = 51, M = 21.95, SD = not reported. EXPERIMETNAL_post_: *n* = 51, M = 14.04, SD = not reported. Post-training also had improved performance on the GOALS (heart rate) for both tasks, *p* < 0.05. Means and SDs are shown pre- and post-for-one-tasks (FT). EXPERIMENTAL_pre_: *n* = 51, M = 94.96, (SD = 11.14). EXPERIMENTAL_post_: *n* = 51, M = 92.71, (SD = 11.67).	2 (performance measures -GOALS) (performance skills − time)	+ Performance skills + Performance measures	Positive
55. [129] ([129])	United States	VR + Teacher + Practice	Med students- infant respiratory distress	168 medical studemmininntmiminns	1 (Teacher)	Medical students completed standard respiratory distress training using traditional didactic material, as well as a mannequin simulation. A randomized group of students also completed an additional 30 min training using immersive virtual reality with various infant simulations (no distress, respiratory distress, and impending respiratory failure). After training, all students completed a free-response test regarding various video questions, including mental status, breathing, breadth sounds, and vital signs for three different cases: no distress, respiratory distress, and respiratory failure. In addition, the need for escalation of care was assessed in each case.	Students who underwent additional VR intervention showed significantly superior status interpretation across all assessed dimensions and all cases, *p* < 0.01. Means and SDs for one case are shown (no distress). CONTROL: *n* = 90, M = 104 (SD = 61.2). EXPERIMENTAL: *n* = 78, M = 124 (SD = 80). The additional VR status groups also had a higher recognition of the need for increased care, *p* = 0.0004. Means and SDs for one case are shown (respiratory failure). CONTROL: *n* = 90, M = 41, (SD = 45.6). EXPERIMENTAL: *n* = 78, M = 56, (SD = 72.7).	2 (skills knowledge- respiratory) (skills knowledge − level of care)	+ Skills knowledge + Skills knowledge	Positive
56. [131] ([131])	United States	AR	Nursing − stroke	36 nursing students	1 (Practice)	Nursing students were divided into two groups: a mannequin-based simulation only (control) or a mannequin-based simulation with AR. They were assessed for clinical judgment with the LCJR.	The AR group spent less time than the control group in the critical phase of the stroke simulation, *p* < 0.05. CONTROL: *n* = 18, M = 99.57 (SD = 79.00). EXPERIMENTAL: *n* = 18, M = 46.61 (SD = 27.92). AR outperformed the control group on the LCJR for one of three sections (noticing), *p* < 0.05. Means and SDs are only shown in Figure 14 in the original publication.	2 (performance skills − time) (clinical reasoning − LCJR)	+ Performance skills + Clinical Reasoning	Positive

Note: Means (M) or medians and standard deviations (SD) or Interquartile Ranges (IQRs) are reported when available. Confidence intervals are reported in place of SD where available and appear in [ ]. The colors for the type of I-XR are as follows: Light yellow/peach = VR, Light blue = AR, gray = MR, and light green = VR/AR. Colors for Study Evidence are as follows: dark green = positive, lighter green = mixed positive, yellow = no difference, red = negative, lighter red = mixed negative, and gold = mixed. * = anything with + indicates I-XR and control received.

**Table 2 behavsci-15-00468-t002:** Learning Theories Noted and Calculated MERSQI/NOS-E and QUADAS scores.

Author, Date	Learning Theory Mentioned	MERSQI + NOS-E Score (Max 24)	QUADAS Score (Max 11)
[3] ([3])	Situated learning theory	22	10
[5] ([5])	No	21.5	11
[6] ([6])	No	21.5	9
[8] ([8])	Cognitive load	15.5	6
[10] ([10])	No	12.5	4
[11] ([11])	No	21	10
[15] ([15])	NLN/Jeffries Simulation Theory	17.5	7
[18] ([18])	No	21	8
[19] ([19])	Constructive Alignment Theory and Blooms	21	8
[20] ([20])	No	13.5	7
[21] ([21])	No	17.5	8
[22] ([22])	Directed self-regulated learning (DSRL) theory; Simulation-based mastery learning	19	8
[23] ([23])	Deliberate practice theory	19.5	9
[24] ([24])	No	20	8
[25] ([25])	No	17.5	8
[26] ([26])	Cognitive load theory	21.5	8
[29] ([29])	No	19	7
[34] ([34])	No	16.5	9
[36] ([36])	No	16.5	9
[38] ([38])	NLN/Jeffries Simulation Theory	16.5	9
[39] ([39])	No	16	7
[41] ([41])	Cognitive load theory; Blooms	14.5	8
[44] ([44])	No	19	7
[50] ([50])	No	20	3
[53] ([53])	No	15	3
[59] ([59])	No	16.5	9
[60] ([60])	No	21	5
[68] ([68])	No	21	7
[69] ([69])	Cognitive load theory	18.5	9
[70] ([70])	No	14	10
[73] ([73])	No	18.5	10
[76] ([76])	No	20.5	9
[82] ([82])	No	17	3
[85] ([85])	No	20	8
[86] ([86])	Cognitive load theory	17.5	3
[87] ([87])	No	21	6
[88] ([88])	No	21	4
[90] ([90])	No	19.5	8
[91] ([91])	No	20.5	9
[92] ([92])	No	21	4
[94] ([94])	No	21	8
[96] ([96])	No	17	2
[100] ([100])	No	17.5	3
[101] ([101])	Kolb	20.5	4
[103] ([103])	No	20.5	10
[106] ([106])	No	19.5	3
[109] ([109])	No	19	9
[110] ([110])	No	20.5	7
[112] ([112])	Experiential learning theory	19.5	8
[118] ([118])	No	16	7
[124] ([124])	No	15.5	7
[126] ([126])	No	19	7
[127] ([127])	No	19	6
[128] ([128])	Cognitive load theory	18	6
[129] ([129])	No	17.5	9
[131] ([131])	No	19	9

*Note:* Cognitive Load theory is not a typical learning theory; however, it was coded here to include the idea of aiding thinking through offloading work to the environment.

## Data Availability

Data spreadsheets for coding the articles are available from the first author upon request via email.

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
