# Peer review of "Immersive Extended Reality (I-XR) in Medical and Nursing for Skill Competency and Knowledge Acquisition: A Systematic Review and Implications for Pedagogical Practices"

_behavsci, 2025, doi:10.3390/bs15040468_

Round 1
Reviewer 1 Report
Comments and Suggestions for Authors
See the attachment.

Author Response
We appreciate the reviewer's thoughtful feedback and acknowledge that the initial framing may have created some ambiguity regarding the role of pedagogy in our review.
Summarizing the Reviewers’ Comments:
- Revised Title and Focus: To better reflect the scope of our review and address the reviewer's concerns, we have revised the title to: "Immersive Extended Reality (I-XR) in Medical and Nursing Education for Skill Competency and Knowledge Acquisition: A Systematic Review and Implications for Pedagogical Practices". We have also made revisions throughout the manuscript to clarify our focus and emphasize the importance of pedagogical considerations in I-XR implementation. We believe these revisions strengthen the manuscript and provide a more comprehensive and nuanced understanding of the role of pedagogy in I-XR-based healthcare education. We appreciate the reviewer's feedback and believe it has significantly improved the quality of our work.
- Clarifying our Focus: The primary objective of this review was to systematically examine the effectiveness of I-XR training compared to non-immersive approaches in enhancing skill competency and knowledge acquisition among healthcare students. While we did not initially intend to conduct an in-depth analysis of pedagogical approaches, our findings led us to recognize the crucial role pedagogy plays in optimizing the implementation and impact of I-XR in healthcare education.
- "Pedagogical Directives": We acknowledge that the phrase "pedagogical directives" may have been unclear. We used this term to refer to the specific guidelines or principles that inform the design and implementation of I-XR learning experiences. In this review, we sought to identify whether studies explicitly or implicitly referenced learning theories to guide their instructional design. This analysis revealed a significant gap in the theoretical foundation of I-XR implementation, highlighting the need for more explicit consideration of pedagogical principles.
- Operational Definition of Pedagogy: We acknowledge that our initial discussion of pedagogy may have been somewhat abstract. We have revised the manuscript to provide a clearer definition of pedagogy and its various components, including learning objectives, instructional strategies, media elements, and assessment methods. We have also elaborated on how these components can be specifically addressed in the context of I-XR.
- Framework for Embodied Learning: To further address the reviewer's concerns, we have now included a comprehensive discussion of an embodied learning framework in the revised manuscript. This framework emphasizes the importance of active engagement, sensory immersion, and interaction with the environment in enhancing learning. We believe this framework provides a valuable tool for educators and developers seeking to optimize the pedagogical effectiveness of I-XR in healthcare education.
- Recommendations for XR Developers: We have also expanded the discussion to include specific recommendations for XR developers. These recommendations highlight the importance of collaborating with educators to ensure that I-XR technologies are designed and implemented in a way that aligns with sound pedagogical principles.
- Affordances of I-XR: We appreciate the reviewer's emphasis on the concept of "affordances." We have incorporated this concept into our discussion of the embodied learning framework, specifically highlighting the importance of bodily adaptation, awareness of constraints, and creative engagement with the affordances of the virtual environment.
- Classification of I-XR Technologies: We acknowledge that the classification of I-XR technologies in Table 1 could be more fine-grained. We have revised the table to include more specific information about the type of technology used in each study, as well as the country of origin.
- Impact of I-XR vs. Non-Immersive technologies: We have reviewed the text to ensure that the distinction between immersive and non-immersive technologies is clear throughout the manuscript, particularly in the discussion of meta-analyses.
- Linking XR Tools to Educational Theories: We agree with the reviewer's point about the need to link XR tools to educational theories. We have expanded our discussion of learning theories and their implications for I-XR implementation. We have also clarified our coding of learning theories, noting that we included both explicit mentions and implicit applications of theoretical principles.
- Optimistic Tone and Robustness of Findings: We have revised the discussion section to moderate the optimistic tone and acknowledge the limitations of the evidence base. We have also clarified that the effectiveness of I-XR may vary depending on the specific outcome being measured.
- Limitations of I-XR Implementations: We have expanded the limitations section to include a broader discussion of the challenges associated with I-XR implementation, including cost, accessibility, and the need for further research on pedagogical approaches.
Reviewer 1 Specific Comments
Major comments:
1/2. This review of I-XR is extremely since its development has led to more and more applications, not always linked to related evaluative research. The lack of evaluative research is critical to be able to validate claims about its impact on learning and development. I also appreciate the focus on deriving “pedagogical directions”….
We thank the reviewer for their kind words and enthusiasm for the current topic.
3. Key in this review is the directives that guide the actual analysis and synthesis of the review literature. This implies that we did look for a clear framework to structure the nature of ‘pedagogical directives’. What is meant by this?
We have broken this large comment down into subsequent parts, but the majority of the concern seems to focus on assuming the purpose of the paper is to investigate the pedagogy involved in influencing the efficacy of the technology on this topic. The primary focus of the paper, however, was to investigate the efficacy of I-XR (compared to non-immersive) technologies in the efficacy of student skill acquisition and knowledge. An additional focus was to review whether learning theory was reviewed or considered in the design to underscore the lack of pedagogy in how healthcare instructors are implementing technology when they are evaluating said outcomes.
We comment on each aspect of the longer comment below. (Please accept that we corrected some grammatical and formatting issues in summarizing the content, as the pdf did not allow for cut and paste).
3a. The current version of the article is hardly satisfactory in answering this key guiding question. A ‘pedagogy’ guides the design and implementation of an instructional approach and defines as such key objectives, the learning content, instructional strategies (that invoke specific learner behavior), the choice of media elements, evaluation and assessment and implies finally organization decisions.
We apologize that the use of the word “pedagogy” in the title might be misleading. The primary focus of the paper was to investigate the efficacy of I-XR (compared to non-immersive) technologies of student skill acquisition and knowledge. An additional focus was to review whether learning theory was reviewed or considered in the design to underscore the lack of pedagogy in how healthcare instructors are implementing technology when they are evaluating said outcomes. Therefore, the paper was not meant to evaluate learning theory or pedagogy approaches directly, but rather point out that most studies evaluating healthcare students’ learning (i.e. skill acquisition and knowledge), seldom reference learning theories when considering their clinical outcomes. However, pointing out this gap of pedagogy in using I-XR technology, we have elaborated on the importance of future work to incorporate this. In doing so, we have now presented a framework of “embodied learning” in the Discussion. Therefore, we have changed the title to “Immersive Extended Reality (I-XR) in Medical and Nursing Education for Skill Competency and Knowledge Acquisition: A Systematic Review and Implications for Pedagogical Practices”. We also believe that we have made this point clearer in the rationale for the current study.
3b. The authors are only somewhat “operational” in that they understand as a ‘pedagogical approach’ in line 50 and further on p. 2……Does this imply that the word pedagogical only refers to learning outcomes? On p.1, line 27 we also see a specific use of the word pedagogical…… But also in this paragraph, the word “pedagogical” remains abstract and is not helpful to direct the analysis and synthesis of the literature.
Thank you for pointing out how pedagogy is defined. With respect to the first instance cited, Baskaran et al. (2023) showed that simulation-based learning (which I-XR entails) includes three important parts. We included that definition because it described the purpose of simulation in skill execution and in exteroceptive awareness (something we elaborate upon in the embodied learning framework). This is not an operational definition for our work.
We do not believe that we are making the claim that pedagogy only includes skill outcome; rather this concern goes back to the nature of the paper. This paper operationalizes “effectiveness” as skill competency and knowledge acquisition. A secondary goal was to address whether pedagogy and/or learning theory was addressed in the studies reviewed.
3c. On p 4., line 153, we read ….. This sentence fits partially our orientation on a framework, and model since theories stress how learning works, how features and then tools influence specific processes and variables.
Thank you; we agree.
3d. They also stress an explicit choice of researchers in their selection of key variables and processes that could be linked to instructional design decisions about objectives, and learning content... We therefore look forward to reading about “theories” in the article. This is being developed but not really translated into concrete list of “directives”.
Thank you for pushing us to expand on a framework for teaching through simulation using I-XR methods. We present a framework of “embodied learning” in the revised paper. To this end, we also include specific recommendations for best practices and how developers and teachers can use this theory to include pedagogy.
3e. What choices should XR developers make in the future? Of course, we do expect a focus on the outcomes of IXR implementations as “part” of a discussion of pedagogy, But is this the only “directive”?
As alluded to above, we offer a specific framework and recommendations for practice in the new, elaborated discussion.
4. The title is promising because it puts emphasis the “immersive” nature of IXR. The focus on this particular “affordance” makes it particularly promising to deal with medical and nursing education. We explicitly use the concept of “affordances” since this could also be an angle to look at the “pedagogical directives” that could be derived from the literature. What specific characteristic/affordance of IXR is key to fit the demands of nursing and medical education and learning? The authors do not do this in a systematic way. … Also section 1.1. on pg. 2 could belong to this kind of analysis since it focuses on especially the unique features of IXR; e.g. Features such as giving students “practical experience and readiness for actual scenarios and procedures”. A solution to remediate this shortcoming is to change the title and to focus solely on the outcomes of IXR. Otherwise, the authors have to restart and redevelop part of the analysis of the selective literature to reach a sound interpretation of what they present as “pedagogical directives”.
Thank you for noting the importance of “affordances”. Indeed, embodied learning, derived from 4E cognition, has a strong emphasis on Gibsonian affordances. To this end, please find that one of the key elements of our newly presented framework includes affordances. By this, we mean the ability to demonstrate a strong understanding of bodily adaptation and awareness of constraints. This could include actively engaging in activities that maximize the use of one’s own bodily design, and creatively adapting to any limitations or constraints. Although in this current paper, we do not go into detail on each of the embodied levels of criteria for this new framework, an accompanying manuscript outlines how this can be done in detail (in preparation). As mentioned previously, given the dearth of pedagogy mentioned in evaluating the efficacy of skill competency and knowledge acquisition, the inclusion of this framework could now be used to evaluate the reviewer’s interest.
Specific Comments:
5. p. 2, line 70. The authors rightly stress that XR is a container concept….But, the article does not apply this in the specific literature review being presented (see table from p. 7). The IXR classification is as such not sufficiently fine-grained since education does not only depend on the nature of the tool, but also on its actual implementation.
Thank you for pointing this out. We appreciate the recognition that IXR does encompass many different, sometimes overlapping, technologies. Table 1 includes whether the study is AR, VR, MR. We have now included a statement in the results in which each is evaluated independently (rather than overall) on the outcomes of students’ skill competency and knowledge acquisition.
6. The article starts with a conceptual base and a first analysis of the available evidence about the “impact” of technologies. We read about VR, AR, and MR but do not always know whether the impact of IXR mirrors the dichotomy between immersive and non-immersive technologies. Could the authors check the text and be sure that this is clear for each meta-analysis mentioned?
We have included the type of technology in the literature review.
7. We know from educational theory reviews (see Hattie and colleagues) that the learning impact is rather related to the implementation approach of tools. This is rightly respected by the authors on p3. Lines 134 and further. This is linked to the need to link the XR-tools to educational theories. On p. 4, the authors rightly stress there is a lack of focus on theoretical embedding of the tools in the teaching and learning process. We applaud the conclusions on p. 4, lines 154 and further that state….. It is therefore it is a pity that this is not further elaborated in terms of “how” these theories could influence the implementation. As stated earlier, this could be influencing the design and implementation of instructional design elements, such as the selection of learning objectives, content, instructional strategies, media characteristics, assessment and evaluation and organizational features. We miss this operational translation of the learning theories focus. Further on, in the table, we find labels of theories…Cognitive load theory. We don’t qualify this as an educational theory since it mainly focuses on cognitive process variables that could be related to a (further not clarified) instructional design choice invoking or inhibiting a type of cognitive load. Other examples are far richer in nature; e.g., experiential learning theory. Key is that the link between these theories and pedagogical directives is underdeveloped.
Thank you for pointing out. Initially, we defined each learning theory, but for the sake of the paper’s length, we chose to omit definitions to allow for a discussion of a newly reposed framework. This framework – embodied learning for IXR- incorporates aspects of constructivist theory and situated learning theory, as they transform a general experiential learning theory (e.g., Kolb) (see Lebert & Vilarroya, 2024). Our new model also builds extensively on Bloom’s theory, in which the later cognitive thinking stages are related to experiential learning through application, analysis, and evaluation.
With regards to Cognitive Load theory, we now point out this is not an educational style. We keep it in the table, as it attests to the idea that “offloading” cognition to the environment – another key aspect of 4E cognition – was considered in these studies.
8. We applaud the adoption of the PRIMSA guidelines…
Thank you.
9. Building on our critical remarks about “pedagogical directives”, we find the current version of the data extraction insufficient. Coding themes do not mirror a rich interpretation of what is to be understood as pedagogical directives… The later is very crude and is done with yes/no or the label found in the articles. No further analysis is carried out. We assume that also articles without an explicit theory label, build on educational theory elements. This is now neglected.
While it is true that we simply coded whether a learning theory was discussed in each paper, this comment reflects early comments that we believe show confusion about the purpose of the paper. The primary purpose of the paper was not to evaluate learning theory. Rather, pointing out this gap of pedagogy in using I-XR technology, was to show that there is a need for learning theory to be applied in these fields. We have also added a sentence clarifying that we only coded whether a learning theory was explicitly mentioned or referred to. This was not limited to formal names, but also included a discussion of any learning theory.
To help with this, we now present a framework of “embodied learning” that could help do so. Again, we apologize if the title implied that we were evaluating learning theory or pedagogy.
10. The analysis table is developed in a consistent way.
Thank you.
11. We applaud the focus on article quality and bias and the fact this is based on standard approach. But we don’t agree with the statement that this is helpful to judge the educational quality of the articles. Rather we consider this related criteria to be methodological in nature.
We apologize for including the word “educational” here. We have removed the word “educational” in line 288. The reviewer is correct in that the MERSQI and QUADAS are ways to assess article quality.
12. The description of the results is mostly in line with the coding categories proposed.
As no further information is given, we assume this is sufficient.
13. The opening sentence of the discussion section (p. 49 lines 373) is very optimistic….Also the conclusion about the robustness of the findings is too optimistic... We urge the authors to be more careful, since the reported evidence is not about the same type of outcomes (compare e.g., cognitive and non-cognitive outcomes). Much depends on the nature of the outcomes being studied. And- when adapting the same perspective of the authors – the majority of the studies does NOT report positive on conclusive outcomes (57,9 M%). We link this to the sentence later on….The IXR based learning environments do as such not deliver consistently what is being promised.
We have changed the first sentence noted in the comment to be less optimistic. However, we disagree that the conclusion about the efficacy of I-XR methods (compared to non-immersive methods) is NOT warranted. There was nearly a 12x increase in the number of studies (43%) between those studies that showed “positive” support for IXR methods compared to non-immersive methods (3% showed that non-immersive were superior). However, it is correct that there was a substantial number of studies that showed “mixed evidence”, depending on the individual outcome evaluated (27%).
14. But we agree with the conclusion on p. 49, line 386…Suddenly the authors present a pedagogical directive that could be related to choice of theory: the selection of “strategies”.
Thank you for the first comment; I think this speaks to the point above.
We are confused by the second half of this comment. 75% of the studies mentioned no learning theory (and when cognitive load was removed), this increased to 84%. Of these, the percentage of study outcomes (Positive, favoring I-XR; Negative, favoring non-immersive; Mixed, etc.) did not differ whether a learning theory was mentioned. We have restated this claim more clearly.
15. p. 49, line 404. The discussion section should mirror the critical remarks state above the conceptual/theoretical base adopted by the authors. It is also a pity that in the context of the review, the actual limitations of IXR implementations were not collected. The limitation mentioned in the discussion section are also limited to psycho-physiological processes and variables and not to education limitations…
Yes, this is true. We feel that it is important in a review of the efficacy of IXR methodology that limitations regarding their cost, implementation, and the like are also acknowledged. The discussion (prior to the limitations) focuses on how the lack of identifiable learning theory. However, we have included this point in the limitations section, as well.
16. Based on the above remarks, the authors could also consider to rewrite the future directions…
Thank you; we have included a newly proposed framework for embodied learning which we feel could be used as a learning theory for pedagogical directives for this type of research.
Reviewer 2 Report
Comments and Suggestions for Authors
The paper selects a valuable research question and aims to examine the effectiveness of I-XR training for healthcare students (e.g., medical and nursing students). The introduction is well written. However, I have major concerns about the analysis and results.
Major comments:
- My primary concern is whether the author can conduct a meta-analysis based on the existing evidence rather than simply presenting the success rates of clinical skills outcomes for different types of VR. If a meta-analysis is beyond the scope of the study or cannot be conducted for specific reasons, these should be clearly explained in the article.
- Beyond discussion on the limitation of application of VR technology in medical education, the study’s limitations should be thoroughly discussed.
Other minor comments:
- Table 1 can be presented in landscape orientation to enhance readability.
- Table 1 can include information about the study country to illustrate the distribution of study areas in the existing literature.
Author Response
We appreciate the reviewer's thoughtful feedback and acknowledge that the initial framing may have created some ambiguity regarding the role of pedagogy in our review.
Summarizing the Reviewers’ Comments:
- Limitations of I-XR Implementations: We have expanded the limitations section to include a broader discussion of the challenges associated with I-XR implementation, including cost, accessibility, and the need for further research on pedagogical approaches.
- Performing a Statistical Meta-Analysis: We are not prepared to conduct a formal statistical meta-analysis. We have included the means, standard deviations, and confidence intervals for each study we reviewed (when possible) so that others may easily use this information to compute Hedges’ G.
- Table 1 arrangement and inclusion of Country: We have requested that the table be reformatted, and now added a column to include the country of origin.
Reviewer 2 Specific Comments
The paper selects a valuable research question and aims to examine the effectiveness of I-XR training for healthcare students (e.g., medical and nursing students). The introduction is well-written. However, I have major concerns about the analysis and results.
Major comments:
- My primary concern is whether the author can conduct a meta-analysis based on the existing evidence rather than simply presenting the success rates of clinical skills outcomes for different types of VR. If a meta-analysis is beyond the scope of the study or cannot be conducted for specific reasons, these should be clearly explained in the article.
Thank you for the suggestion. The goal of this paper was to provide a descriptive summary of the efficacy of I-XR methods, and while a statistical meta-analysis would provide a quantitative measure to address this question, at this time we are not prepared to conduct a formal statistical meta-analysis. We have included the means, standard deviations, and confidence intervals for each study we reviewed (when possible) so that others may easily use this information to compute Hedges’ g.
2. Beyond discussion on the limitations of application of VR technology in medical education, the study’s limitations should be thoroughly discussed.
Thank you for pointing this out. We have expanded on the limitations of this study and its generalizability while offering a new framework. To that end, we have minimized discussion of inherent limitations of I-XR, as there are plenty of good reviews that do so.
Other minor comments:
1. Table 1 can be presented in landscape orientation to enhance readability.
We have suggested to the formatting editors that this would enhance readability.
2. Table 1 can include information about the study country to illustrate the distribution of study areas in the existing literature.
We have included the country of origin.
Round 2
Reviewer 2 Report
Comments and Suggestions for Authors
The authors have addressed the reviewer's comment.
1) The readability of Table 1 could be enhanced by adjusting its formatting (e.g., aligning columns consistently, using bold headers, or adding color contrast for key data points). 2) Similarly, Figure 8 would benefit from improved labeling (e.g., clarifying category abbreviation as notes (e.g. AR)). Notably, only one study on VR/AR and one on MR were included in the analysis, and both reported outcomes categorized as 100% positive change/ or 100% no difference change. This limited sample size highlights a critical gap in the literature and underscores the need for caution in generalizing these results.
Author Response
We appreciate the reviewer's additional comments. We appreciate the time to improve on this version.
Summarizing the Reviewers’ Comments
Please note that we have asked the journal to re-format Table 1 to landscape for better readability. We have included the percentages of AR vs. VR for each of the assessment categories in the results and in the discussion. We noted that the inclusion of only one MR study and one study that was labeled as AR/VR (because we could not determine the specific methodology) are limitations (see p. 17-18). Finally, we included descriptions to Figure 8.
Reviewer 2 Specific Comments
1) The readability of Table 1 could be enhanced by adjusting its formatting (e.g., aligning columns consistently, using bold headers, or adding color contrast for key data points).
We have included colors for the type of technology and the final study assessment (Positive, Negative, Mixed, etc.). We have placed the date with the authors, removing one column, and we have also reformatted the width of the longest columns to be able to allow for fewer pages of the table. We have asked the journal to re-format Table 1 to landscape for better readability.
2) Similarly, Figure 8 would benefit from improved labeling (e.g., clarifying category abbreviation as notes (e.g. AR).
We have added in descriptions of the final assessments (Positive, Negative, etc.), and the percentages in a data table within the figure.
3). Notably, only one study on VR/AR and one on MR were included in the analysis, and both reported outcomes categorized as 100% positive change/ or 100% no difference change. This limited sample size highlights a critical gap in the literature and underscores the need for caution in generalizing these results.
We noted the limitations of only one MR study and one AR/VR study (because we could not determine the specific methodology) are limitations (see p. 16-17 and 18).
